



# Meteorology-normalized impact of COVID-19 lockdown upon NO$_2$ pollution in Spain

Hervé Petetin[1], Dene Bowdalo[1], Albert Soret[1], Marc Guevara[1], Oriol Jorba[1], Kim Serradell[1], and Carlos Pérez García-Pando[1,2]

[1]Barcelona Supercomputing Center, Barcelona, Spain
[2]ICREA, Passeig Lluís Companys 23, 08010 Barcelona, Spain

**Correspondence:** Hervé Petetin (herve.petetin@bsc.es)

**Abstract.** The spread of the new coronavirus (COVID-19) forced the Spanish Government to implement extensive lockdown measures to reduce the number of hospital admissions, starting on March 14$^{th}$ 2020. Over the following days and weeks, strong reductions of nitrogen dioxide (NO$_2$) pollution were reported in many regions of Spain. A substantial part of these reductions is obviously due to decreased local and regional anthropogenic emissions. Yet, the confounding effect of meteorological variabil-

ity hinders a reliable quantification of the lockdown impact upon the observed pollution levels. Our study uses machine learning (ML) models fed by meteorological data along with other time features to estimate the "business-as-usual" NO$_2$ mixing ratios that would have been observed in the absence of the lockdown. We then quantify the so-called meteorology-normalized NO$_2$ reductions induced by the lockdown measures by comparing the business-as-usual with the actually observed NO$_2$ mixing ratios. We applied this analysis for a selection of urban background and traffic stations covering the more than 50 Spanish

provinces and islands.

The ML predictive models were found to perform remarkably well in most locations. During the period of study, going from the enforcement of the state of alarm in Spain on March 14$^{th}$ to April 23$^{rd}$, we found the lockdown measures to be responsible for a 50% reduction of NO$_2$ levels on average over all provinces and islands. The lockdown in Spain has gone through several phases with different levels of severity in the mobility restrictions. As expected the meteorology-normalized change of NO$_2$

was found to be stronger during the phases II (the most stringent one) and III than during phase I. In the largest agglomerations where both urban background and traffic stations were available, a stronger meteorology-normalized NO$_2$ change is highlighted at traffic stations compared to urban background ones. Our results are consistent with foreseen (although still uncertain) changes in anthropogenic emissions induced by the lockdown. We also show the importance of taking into account meteorological variability for accurately assessing the impact of the lockdown on NO$_2$ levels, in particular at fine spatial and

temporal scales.

Meteorology-normalized estimates such as the ones presented here are crucial to reliably quantify the health implications of the lockdown due to reduced air pollution .



# 1 Introduction

The rapid spread of the new coronavirus (COVID-19) has led numerous countries worldwide to put their citizens on various
forms of lockdown, with measures ranging from light social distancing to almost complete restrictions on mobility (Anderson
et al., 2020). Spain has been among the countries most severely affected by COVID-19, and where proportional (and therefore
drastic) containment measures have been implemented. Spanish authorities declared the constitutional state of alarm on March
13th 2020, to be enforced on the 14th. During this period (phase I) residents had to remain in their primary residences except for
purchasing food and medicines, work or attend emergencies. Non-essential shops and businesses, including bars, restaurants,
and commercial businesses had to close. Due to the persistent rise in hospital admissions, an even more severe second phase
(phase II) of the lockdown was implemented between March 30th and April 9th, when only essential activities including food
trade, pharmacy, and some industries were authorized. A third phase (phase III) started on April 10th, when some non-essential
sectors, including construction and industry, were allowed to return to work.

The shutdown of both social and economic activities in Spain has reduced anthropogenic pollutant emissions. Among the sec-
tors presumably most affected, road transport, which is a dominant source of air pollution in urban areas, and air traffic have
fallen to unprecedentedly low levels. The impact on the industrial sector is likely more contrasted, as some essential industries
(e.g. fuel and energy related, petrochemical) were authorized to continue their production, while some others were forced to
halt their activity.

While such an extraordinary situation has obviously impacted the levels of air pollution in the country (Tobías et al., 2020),
the extent of such reductions remains uncertain. Besides emissions, air pollution is strongly influenced by meteorological
conditions driving their dispersion and short- to long-range transport, and affecting their removal and chemical evolution. As
highlighted by Tobías et al. (2020) in Barcelona, this makes the quantification of air pollution reductions during the lockdown
unreliable when solely based on the analysis of in-situ observations. Chemistry-transport models (CTMs) are an essential tool
for investigating both actual and alternative states of the atmosphere under different emission scenarios. Actually, the lockdown
offers unique opportunities for so-called dynamical CTM evaluations (Rao et al., 2011), i.e., testing the ability of CTMs to re-
produce the observed changes of concentrations under unusually different emissions (Guevara, in preparation). However, given
the difficulty of accurately estimating the changes in emissions induced by the lockdown along with the inherent limitations
of CTMs, particularly in urban areas, estimating the reductions with this method remains a complex task sullied by substantial
uncertainties that are difficult to quantify.

The need for attributing changes in pollutant concentrations to changes in emissions recently motivated the development of so-
called weather normalisation techniques based on machine learning (ML) algorithms (Grange et al., 2018; Grange and Carslaw,
2019). The idea consists in training ML models to predict pollutant concentrations at air quality (AQ) monitoring stations based
a set of features including meteorological data and other time variables. This allows for the building of ML models that learn
the influence of meteorology upon pollutant concentrations under a given average emission forcing. These ML models can
then be used for predicting pollutant concentrations under a range of meteorological conditions, with the associated average
referred to as meteorology-normalized time series in Grange et al. (2018) and Grange and Carslaw (2019). In addition, such





ML models can be used for predicting business-as-usual pollutant concentrations during periods with presumably different emissions, i.e., estimating the pollutant concentrations that would have been experienced without the change in emissions. Following the ideas introduced in Grange et al. (2018) and Grange and Carslaw (2019), the present study uses ML models
to investigate the reduction of nitrogen dioxide ($NO_2$) concentrations in Spain due to the COVID-19 lockdown. Since road transport and industry are major sources of $NO_2$ emissions, the impact of the lockdown on this primary pollutant is expected to be strong and thus easier to detect and quantify. Due to its short lifetime and relatively simple chemistry, $NO_2$ is likely more directly impacted by meteorological conditions than other pollutants like particulate matter that depend upon more numerous and complex processes.


## 2 Data and methods

### 2.1 $NO_2$ data

This study primarily relies on hourly $NO_2$ measurements performed routinely in Spanish AQ surface monitoring stations. We considered the time period going from 2013/01/01 to 2020/04/23. We used the $NO_2$ data available through the GHOST
(Globally Harmonised Observational Surface Treatment) project developed at the Earth Sciences Department of the Barcelona Supercomputing Center. GHOST is a project dedicated to the harmonisation of global surface atmospheric observations and metadata, for the purpose of facilitating quality-assured comparisons between observations and models within the atmospheric chemistry community (Bowdalo, in preparation). GHOST ingests numerous publicly available AQ observational datasets. In this study, we used the $NO_2$ data from the European Environmental Agency (EEA) AQ e-Reporting (EEA, 2020). We prioritized
the validated data (E1a) and used the near-real time data (E2a) only when necessary. (Nearly all the 2013-2019 $NO_2$ data are E1a and data in 2020 are E2a.) All $NO_2$ measurements taken into account here are operated using chemiluminescence with an internal Molybdenum converter, the dominant European methodology for measuring $NO_2$. GHOST provides a wide range of harmonized metadata and quality assurance (QA) flags for all pollutant measurements. In this study, we took benefit of these flags to apply an exhaustive QA screening. More details on the QA flags used can be found in Appendix A.
$NO_2$ measurements are available over the period 2013 to 2020 in 551 stations in Spain. This study aims at investigating the reduction of $NO_2$ over a variety of environments and geographical locations. We thus designed an algorithm for automatically selecting (when possible) one urban/suburban background station and one traffic station in each Nomenclature of Territorial Units for Statistics level 3 (NUTS-3) (Ceuta and Melilla excluded), which corresponds to Spanish provinces over mainland and individual islands over the Balearic and Canary Islands (hereafter referred to as provinces for convenience). After the QA
screening of $NO_2$ data, we set different thresholds for minimum data availability over different periods of interest, namely 50% of daily data over the entire period of study, 50% over the period 2017/01/01-2019/01/01 (used for training the ML models, see below), 25% over the period 2020/01/01-2020/03/13 (used for testing the ML models) and 10% during the lockdown period. Stations in each province were then selected to maximize the surrounding population density (within a geodesic radius of 5 km) and the data availability (both before and during the lockdown). The population density at AQ monitoring stations was retrieved


through GHOST, which ingests the Gridded Population of the World (GPW) version 4 dataset (Center for International Earth Science Information Network - CIESIN - Columbia University, 2018). Stations fulfilling the different criteria were identified in 50 provinces of Spain and are considered in this study (38 provinces with urban background stations and 37 provinces with traffic stations). No appropriate stations were found in Palencia, Ávila and some islands (La Palma, La Gomera, El Hierro, Lanzarote, Eivissa and Formentera). A map of the entire NO$_2$ monitoring network is shown in Fig. 1 together with the stations

selected in each Spanish province. Names and geographical locations of the stations are reported in Table A1 in Appendix.

## 2.2 Meteorological data

Meteorological data are taken from the ERA5 reanalysis dataset (Copernicus Climate Change Service (C3S), 2017). ERA5 data have a spatial resolution of about 31 km. At all AQ monitoring surface stations, we extracted the following variables at the daily scale : daily mean 2-m temperature, minimum and maximum 2-m temperature, surface wind speed, normalized 10-m

zonal and meridian wind speed components, surface pressure, total cloud cover and boundary layer height.

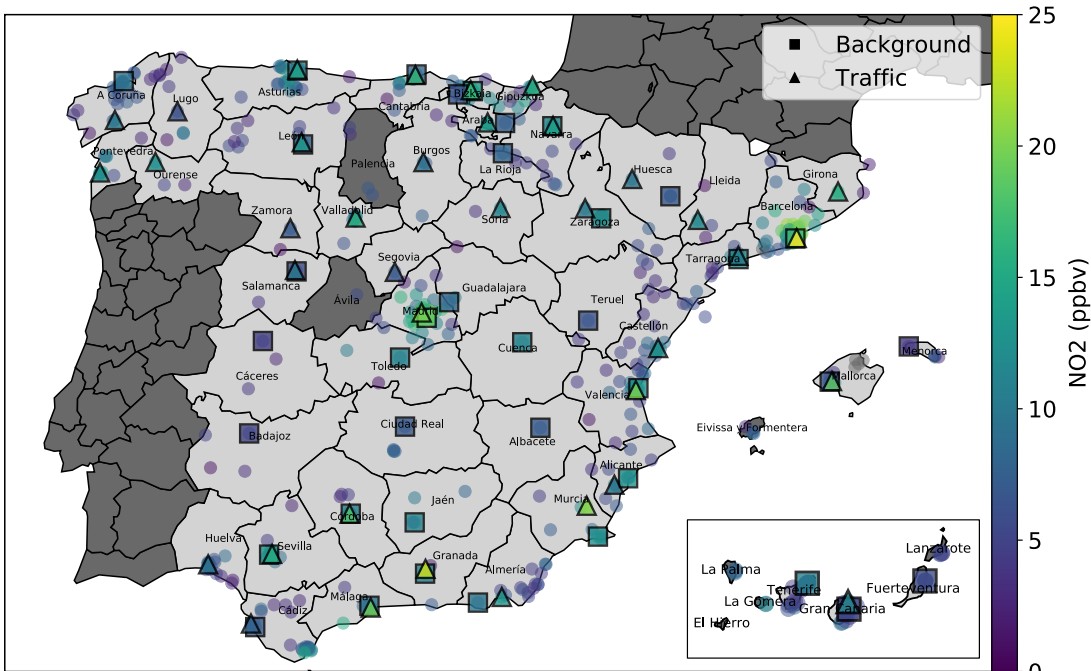

**Figure 1.** Mean NO$_2$ mixing ratios [ppvb] (2013-2020) at all (circles) and selected (squares and triangles) stations. Administrative borders show the NUTS-3 administrative units, which correspond to the Spanish provinces over mainland and to individual islands. Dark gray areas indicate provinces and islands with a lack of stations that fulfill the selection criteria.



## 2.3 Methodology

We implement and train ML models to estimate the daily $NO_2$ mixing ratios that would have been observed without the implementation of the lockdown in each selected station, i.e. under business-as-usual emission forcing. Hereafter, we will refer to these mixing ratios as business-as-usual $NO_2$.

### 2.3.1 Machine learning model


In this study, we retain the Gradient Boosting Machine (GBM), a popular decision tree-based ensemble method belonging to the boosting family (Friedman, 2001). More information on this model is given in Appendix B. ML models based on decision trees offer several interesting attributes. First, they internally handle the process of feature selection, which allows including potentially useless features without strong deterioration of the prediction skills. Second, they provide useful information about the importance of the different features. Third, in contrast to most parametric methods that derive a unique (more or less sophis-


ticated) function supposedly valid over the whole features' space, non-parametric methods based on decision trees internally rely on successive splitting operations (a mother branch being divided into two daughter branches), which may be convenient for designing one single model able to work efficiently under different seasons and weather regimes.

### 2.3.2 Choice of features and modelling strategy


Following the work of Grange and Carslaw (2019), the idea here is to use past recent data to train a ML model able to reproduce the $NO_2$ mixing ratios based on a combination of meteorological features and other time features. The features used in this study are : daily mean 2-m temperature, minimum and maximum 2-m temperature, surface wind speed, normalized 10-m zonal and meridian wind speed components, surface pressure, total cloud cover, boundary layer height, date index (days since 2013/01/01), Julian date and weekday. All the data used in this study are daily. Some pollutant concentrations are known to


strongly vary depending on the season, day of week and hour of the day, notably due to the variability of emissions and chemistry. The two last time features act as proxies for these processes and aim at representing their climatological variations. Over longer (multi-annual) time scales, typically air pollutant concentrations cannot be considered as stationary due to substantial trends (especially in emissions), which is intrinsically problematic for training ML models. Following Grange et al. (2018) and Grange and Carslaw (2019), we introduced the date index as a proxy for this potential trend. Including such a feature with


unique values is not expected to directly help the ML model to learn about $NO_2$ variability. However, it allows us to train one single ML model over a relatively long and thus potentially non-stationary time series. In contrast to linear regression, GBM does not learn equations relating the target variable to the different features, but rather builds non-parametric relationships between target and features. As a consequence, such a model will always make $NO_2$ predictions within the range of $NO_2$ values used in the training, regardless of the inclusion of the aforementioned date index feature or the feature values it takes for


making the predictions. However, if $NO_2$ strongly increases (decreases) with time in the training dataset, the GBM model is able to split the data using the trend feature and therefore predict $NO_2$ in the range of the higher (lower) mixing ratios reached by the end of the training period. We note that even with a trend feature, such a model is not expected to stay valid very far in





time relative to the training data when the training data is following a too strong trend. Our sensitivity tests have clearly shown that the behaviour of the ML models substantially improves when including the trend feature.

In our study, the GBM models are trained over the 3 last full years, namely 2017-2019 and then used for predicting business-as-usual $NO_2$ mixing ratios over the 4 following months, from January to April 2020. Such a duration is expected to allow capturing a substantial part of the inter-annual variability of $NO_2$ mixing ratios and meteorological conditions and ensures some past data is available for quantifying the uncertainties of our ML modelling strategy (as explained later in Sect. 2.3.3). Note that no improvement was found with extended training periods of 4 or 5 years. Although our interest is to predict $NO_2$

during the lockdown period, the two and half preceding months were kept to test the validity of our predictions and uncertainty estimates.

The machine learning modeling in this study is performed using the *scikit-learn* Python package (Pedregosa et al., 2011). The GBM model comprises a number of hyperparameters to be tuned. Since features are temporal variables, instances cannot be considered as independent due to autocorrelation. We thus tuned our ML models using the so-called time series cross-validation

with five splits, which corresponds to a rolling-origin cross-validation in which data used for the validation is always posterior to the data used for the training (*TimeSeriesSplit* in *scikit-learn*). Over a selection of the most important hyperparameters, we applied a so-called *randomized search* over a range of possible hyperparameter values. Compared to the so-called *grid search* in which all combinations of hyperparameters are tested, the randomized approach tests only a certain number (20 in our case) of tuning configurations chosen randomly. This allows to explore a large part of the hyperparameters space at a greatly reduced

computational cost, and tends to be less prone to overfitting. More details on the tuning of the GBM model can be found in Appendix C.

### 2.3.3 Uncertainty estimation

In order to quantify our prediction uncertainty, we replicated four similar experiments over the past years since 2013, i.e., training ML models over 2013-2015, 2014-2016, 2015-2017 and 2016-2018, and testing them over the 4 first months of 2016,

2017, 2018 and 2019, respectively. We obtained on average 538 daily residuals (predicted minus observed $NO_2$ daily mixing ratios) for each station and we took the associated 5[th] and 95[th] percentiles as the uncertainty interval for our ML-based predictions of daily $NO_2$ mixing ratios. Therefore, for each station, we obtained a fixed asymmetric 90% confidence interval used to characterize the uncertainty of our predictions during the first 4 months of 2020. Averaged over all Spanish provinces, the uncertainty interval is [-5.5, +5.2] ppbv at urban background stations, and [-6.7, +6.6] ppbv at traffic stations.

In 2020, the period before the lockdown, namely January 1[st] to March 13[th], is used to check the performance of the ML models trained over 2017-2019 against the observed $NO_2$ mixing ratios, given the aforementioned uncertainty. Ideally, we expect the differences between observed and predicted $NO_2$ mixing ratios to remain within the estimated uncertainty during that period. Conversely, after April 14[th], due to the reduction of $NO_2$ emissions caused by the lockdown, we expect the observed $NO_2$ mixing ratios to quickly decrease compared to the business-as-usual $NO_2$ mixing ratios predicted by the ML model, eventually

down to a level at which the differences are statistically significant.





These uncertainties are suited for our ML-based daily $NO_2$ predictions. Since daily uncertainties are at least partly uncorrelated, predictions averaged over longer time periods are expected to have smaller uncertainties. We estimated the uncertainty affecting our ML predictions at the weekly scale. We used a similar approach than previously described for the daily uncer-

tainty, but based on the 7-day running average of the daily residuals (by requiring a minimum of 5 over 7 days with available data). The 5th and 95th percentiles were computed based on the entire set of residuals (514 residuals on average at each station over 2016-2019). On average over all provinces, the weekly uncertainty interval obtained are [-4.2, +3.5] ppbv at urban background stations, and [-4.9, +4.5] ppbv at traffic stations, which represents a reduction of 27 and 30%, respectively, with respect to the daily uncertainties.

Our main interest in this study is to quantify the mean $NO_2$ changes during the lockdown period. We decided to keep the weekly scale uncertainties for the predictions of business-as-usual $NO_2$ mixing ratios averaged over its different phases (10-13 days each) and over the entire lockdown period (41 days). The use of weekly uncertainties is likely conservative when used for the entire lockdown average, but accounts for potential data gaps, particularly when estimating the shorter phases therein.

## 3   Results and Discussion

In this section, we first evaluate the ML-based predictions of business-as-usual $NO_2$ mixing ratios (Sect. 3.1). We then illustrate our methodology in the two provinces with largest population density, namely Madrid and Barcelona (Sect. 3.2). We then analyze the meteorology-normalized changes of $NO_2$ obtained in all Spanish provinces (Sect. 3.3). We discuss in Sect. 3.4 the potential relationships with emission reductions. Finally, we discuss in Sect. 3.5 the advantages of our ML-based approach for estimating the baseline $NO_2$ pollution compared to the climatological approach.

### 185   3.1   Evaluation of ML predictions

The performance of the ML predictions is shown in Fig. 2, and the statistics averaged over all Spanish provinces reported in Table 1. Results are evaluated using the following metrics, calculated based on daily $NO_2$ mixing ratios : mean bias (MB), normalized mean bias (nMB), root mean square error (RMSE), normalized root mean square error (nRMSE) and Pearson correlation coefficient (PCC).

For information purposes, we included the statistical results obtained over the training dataset (2017/01/01-2019/12/31). Checking results over the training data may be useful for highlighting obvious situations of overfitting, when the performance is almost perfect. At urban background stations, results show no bias, low RMSE (always below 30%, 20% on average over all provinces), and a high PCC (0.91 on average). Similar results are obtained at traffic stations. Although such a performance obtained is very good, there are no clear signs of too prejudicial overfitting at this stage.

On the test dataset (2020/01/01-2020/03/13, before the lockdown), the performance remains reasonably good in most provinces. On average over the different provinces, the bias increases to +2 and +7%, the RMSE to 32 and 28%, and the PCC is reduced to 0.71 and 0.75, at urban background and traffic stations, respectively. Results highlight some inter-regional variability of the performance, with poorer statistics in some provinces, at least for one type of station. At most stations, the bias remains below





**Table 1.** Performance of the ML predictions of NO$_2$ mixing ratios on the test dataset (2020/01/01-2020/03/13), averaged over all Spanish provinces (the standard deviation among the provinces is also indicated).

| Dataset | Type of station | MB [ppbv] (nMB [%]) | RMSE [ppbv] (nRMSE [%]) | PCC | N |
|---------|-----------------|---------------------|-------------------------|-----|---|
| Training | Urban background | -0.0±0.0 (-0%) | 1.8±0.7 (20%) | 0.91±0.06 | 957 |
| Training | Traffic | +0.0±0.0 (+0%) | 2.4±1.0 (18%) | 0.91±0.05 | 990 |
| Test | Urban background | +0.1±1.5 (+2%) | 3.4±1.2 (32%) | 0.71±0.11 | 62 |
| Test | Traffic | +0.8±1.9 (+7%) | 4.0±1.5 (28%) | 0.75±0.10 | 66 |

±20% while nRMSE ranges between 15 and 45% (highest nRMSE around 50% in Teruel and Tenerife). Most provinces show
PCC around 0.6-0.9, with only a few exceptions below 0.6 (urban background sites in Bizkaia, Fuerteventura, Huesca, La Rioja and traffic site in Granada and Gran Canaria).

Several factors may explain the poorer statistical results obtained at some stations. First and foremost, it may be due to deficiencies in the ML modelling, and in particular to some overfitting. This seems to be the case of Fuerteventura and Huesca, given the good performances obtained with the training data. Since we are considering numerous stations in this study, we need
a fixed procedure applied similarly to all ML models to be trained. As described in Sect. 2.3.2, we designed our training and tuning procedure in order to limit as much as possible this common issue, through rolling-origin cross-validation and randomized search in the hyperparameters space. Overall results are satisfactory but some overfitting can still persist in some cases.

Second, although moderately, some of the biases and errors may be partly due to trends and/or inter-annual variability of NO$_2$. As previously explained (Sect. 2.3.2), by model design, if NO$_2$ levels in the first months of 2020 are outside of the NO$_2$ range
in the 2017-2019 training dataset, our predictions over the lockdown period could be equally biased. The different NO$_2$ time series indeed show some cases where NO$_2$ mixing ratios are lower than in the past years (since 2013). In the frame of our study, it is important to mention that, although the lockdown was officially implemented on March 14$^{th}$, the COVID-19 started to perturb the business-as-usual situation in the days/weeks before, first through the cancellation of numerous events and, later, through unusual movements of a part of the population (e.g. to second homes). Although complicated to assess more precisely
in each of the Spanish provinces, this likely explains part of the biases noticed in the second half of the test period.

Third, poor performances at some stations may be due to weaker relationships between meteorological input data and NO$_2$ mixing ratios. This points to uncertainties in the ERA5 meteorology data. For example, the relatively coarse spatial resolution (31 km) of ERA5 data may only capture part of the meteorological variability existing at a given station. This is especially true when considering stations located in urban areas where the complex urban morphology (e.g. presence of buildings, canyon
street) is known to locally distort the mesoscale circulation. Decision-tree based ML methods like GBM offer some interpretability by providing a measure of the importance of the different features included as input data. In our case, on average over all ML models, the most important feature is the boundary layer height (22±7%) followed by the surface wind speed (12±5%). These two parameters drive the ventilation and dispersion of the pollutants emitted at the surface, and their variabil-



ity at some stations may be only partly captured by the ERA5 data at some urban stations. Also, the ERA5 data may poorly

capture the meteorological conditions in some stations located on small islands with complex orography, like in the Canary islands (e.g. Tenerife and Fuerteventura).

The chosen training and tuning procedures applied in this study were designed to limit some of these different sources of uncertainty, but persistent errors cannot be excluded. This is why we added another layer of analysis in which we estimated

the uncertainties of our ML predictions by replicating exactly the same procedure over the past years since 2013 (as explained in Sect. 2.3.3). Computed as the 5$^{th}$ and 95$^{th}$ percentiles of the daily residuals obtained over a large test period extending over several years (2016-2019), the uncertainty intervals are expected to cover most (90%) of the errors caused by these different sources of uncertainties. Indeed, considering all stations, our results indicate that 88% (3966 points over 4523) of the $NO_2$ observations in 2020 before the lockdown fall within the corresponding prediction uncertainty interval at each station, thus

very close to 90%. This demonstrates that the daily uncertainty estimated in this study is well quantified.

All in all, we have shown that our ML predictions and associated uncertainties are qualified for estimating the business-as-usual $NO_2$ mixing ratios during the lockdown.

### 3.2 Illustration of the results in Madrid and Barcelona

#### 3.2.1 Madrid

The daily $NO_2$ mixing ratios observed and predicted in the province of Madrid are shown in Fig. 3 for both the urban background station and the traffic station, with station codes-*names* ES1941A-*Ensanche de Vallecas* and ES1938A-*Castellana*, respectively. The $NO_2$ mixing ratios observed over the past years since 2013 are also included. Since days of week are not consistent from one year to the other, we also show the $NO_2$ 7-day running mean time series where a minimum of 5 over 7 days is required to compute the average.

In Madrid, the ML reproduces remarkably well the variability of $NO_2$ mixing ratios at the urban background and traffic stations before the lockdown (nMB of -4 and +3%, nRMSE of 19 and 21%, PCC of 0.87 and 0.84, respectively). Importantly, prediction errors remain within the uncertainty interval. The two sub-periods with lower $NO_2$ mixing ratios, during the second half of January and early March occur concomitantly with strong wind speeds in Madrid, above 6 m s$^{-1}$ on a daily average (above the 95$^{th}$ percentile of the ERA5 daily wind speed over 2013-2020 during this season), and relatively high boundary layer heights

(up to 1000-1500 m on a daily average). It is worth mentioning that a low emission zone (LEZ) with relatively strict vehicle restrictions applied for entering a limited area of about 5 km$^2$ corresponding to the heart of the city center was implemented in early January 2020. Such a change in emissions may in principle directly impact the performance of the ML predictions by inducing a positive bias (since the ML models are designed precisely for highlighting such events). In our case, considering that the LEZ was still in its transition phase (offending motorists were not yet receiving fines) and that the two stations selected

in Madrid province are located outside this LEZ (at 9 and 3 km from the city center), a limited impact is expected.

After the implementation of the lockdown, the observed $NO_2$ mixing ratios decreased down to about 11 and 7 ppbv on aver-

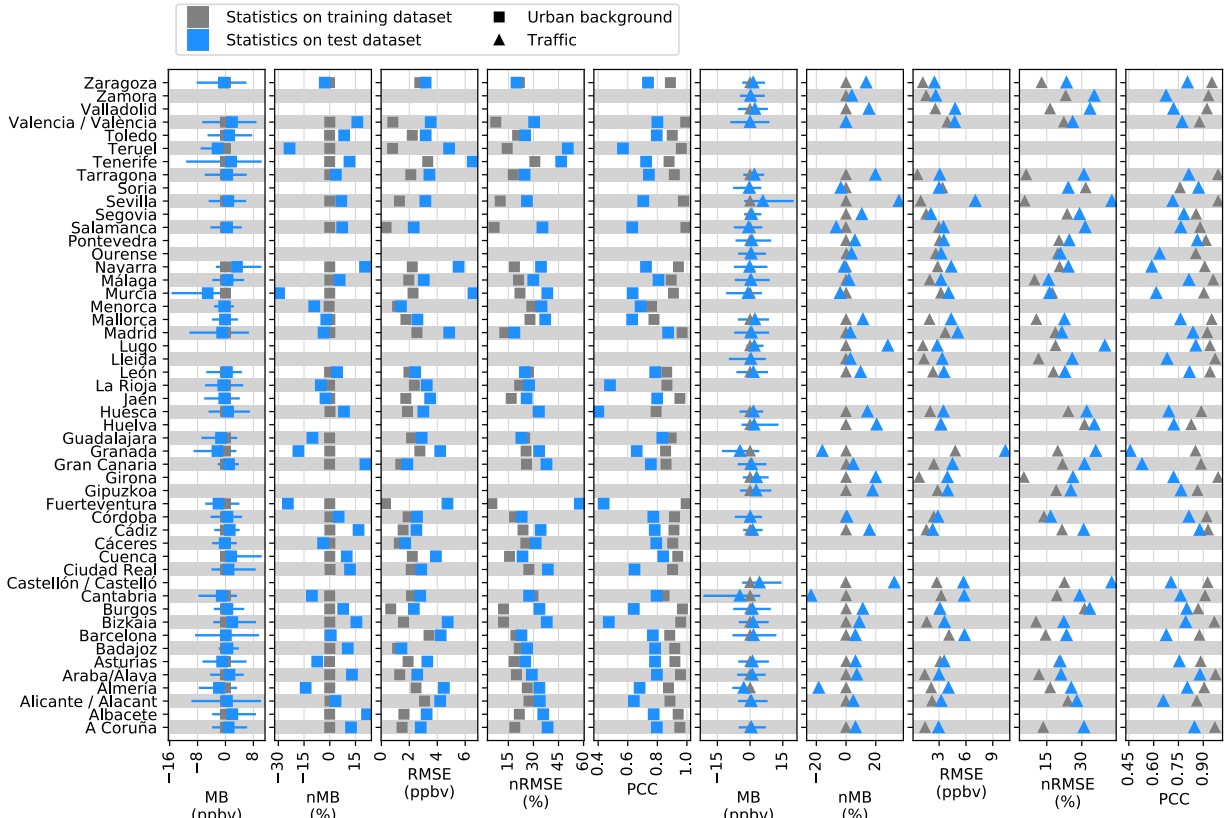

**Figure 2.** Statistical results of the ML-predicted business-as-usual NO$_2$ mixing ratios over training dataset (2017-2019, in grey) and test dataset before lockdown (2020/01/01-2020/03/13, in blue). Metrics are mean bias (MB), normalized mean bias (nMB), root mean square error (RMSE), normalized root mean square error (nRMSE), Pearson correlation coefficient (PCC) and number of points (N, only shown for the test dataset). The horizontal bars added to MB correspond to the uncertainty (90% confidence interval) at the daily scale.

age, and reached daily minimum mixing ratios of 6 and 3 ppbv, respectively, over the entire period. Compared to the previous years, the NO$_2$ mixing ratios at the urban background site are clearly in the lower tail of the distribution. In the traffic site, never NO$_2$ levels had been so low for such an extended period of time at least since 2013. In comparison, business-as-usual

NO$_2$ mixing ratios at these two sites would have remained around 17-18 ppbv on average. After the lockdown, the differences between the observed and business-as-usual NO$_2$ are found to progressively increase, becoming more and more statistically significant. This demonstrates unambiguously that the lockdown considerably reduced the NO$_2$ pollution in Madrid, regardless of the meteorological conditions, which points to a drastic decrease of the business-as-usual emission forcing.

We computed the meteorology-normalized change of NO$_2$ during the lockdown period covered by this study (from March

14[th] to April 23[th]) as the mean difference between ML-based business-as-usual and observed NO$_2$ daily mixing ratios. The uncertainty at weekly scale is here used as an estimate of the uncertainty (at 90% confidence level) affecting the mean NO$_2$





change. On average over the entire lockdown period, NO$_2$ levels have decreased by -7[-12,-1] ppbv at the urban background station, which corresponds to -40[-66,-3]% in relative terms. The impact is faster, stronger and more statistically significant at the traffic station than in the urban background one, with a mean NO$_2$ reduction of -9[-14,-4] ppbv, or -56[-86,-24]% in

relative terms. This result is consistent with a lockdown affecting most strongly the sector of traffic emissions. At the daily scale, the reduction of NO$_2$ in Madrid reached its maximum at the end of the second and more stringent lockdown phase, while a strong reduction persisted during the third phase.

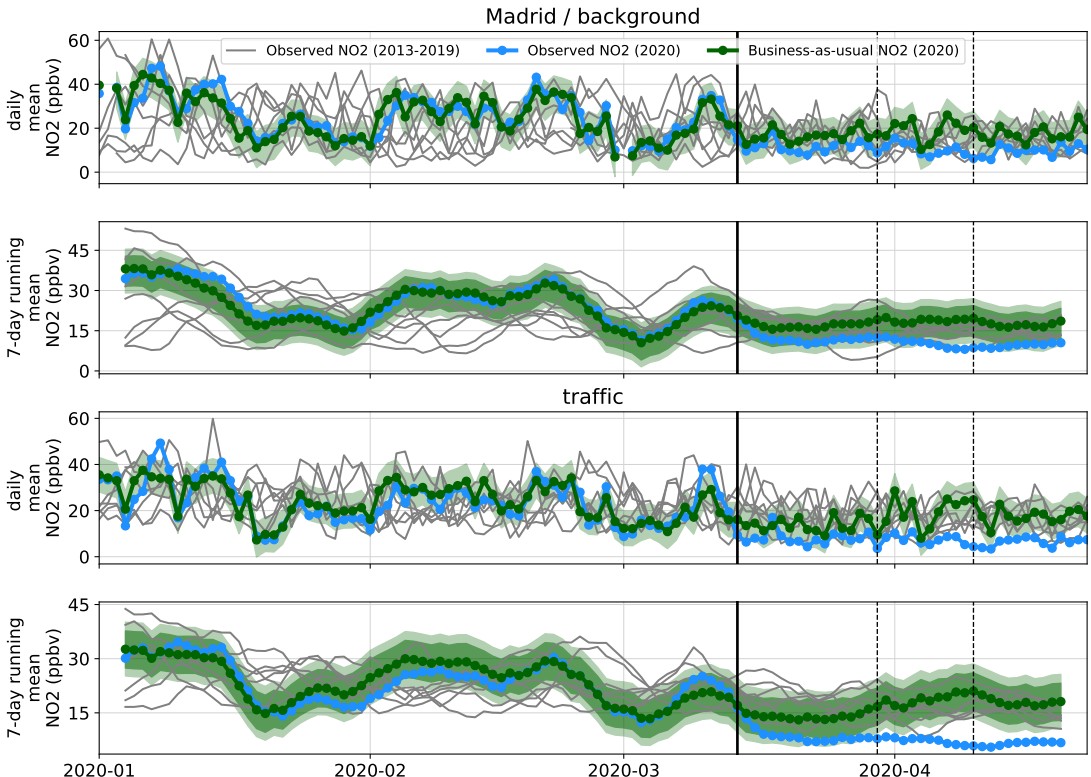

**Figure 3.** NO$_2$ mixing ratios in Madrid province. The two top panels show the daily mean and 7-day running mean at the urban background station, respectively. The two bottom panels show the time series at the traffic station. Each panel displays the NO$_2$ mixing ratios observed in 2020 (in blue) and during the past years (2013-2019, in grey), and predicted in 2020 by the ML model (in green). The uncertainties of the ML predictions are given at a 90% confidence level at the daily (light green) and weekly scales (medium green). The vertical black line identifies the beginning of the lockdown, the next dotted lines separate the different lockdown phases (phase I : 2020/03/14-2020/03/29, phase II : 2020/03/30-2020/04/09, phase III : 2020/04/10-2020/04/24).





### 3.2.2 Barcelona

Figure 4 presents the results in Barcelona for both the urban background and traffic stations, with station codes-*names*
ES1396A-*Sants* and ES1480A-*L'Eixample*, respectively. Compared to Madrid, the ML predictions in Barcelona have rela-
tively similar errors (nRMSE of 23%) and correlations (PCC of 0.77 and 0.82, respectively). The bias is very low at the urban
background station (+1%), and reaches +12% at the traffic station, which largely remains within the uncertainty interval. The
positive bias in the traffic station started in late January and persisted during the following weeks, particularly after the second
week of February. The ML model failed at reproducing these low $NO_2$ mixing ratios notably because some of the observed
$NO_2$ mixing ratios during that period were lower than during the previous years. As in Madrid, a LEZ was implemented in
Barcelona, starting in early January 2020, with less stringent vehicle restrictions but over a larger area (95 km$^2$). Both the
urban background and traffic stations selected in Barcelona are included in this LEZ. The potentially stronger effect of the LEZ
at traffic stations could explain at least partly this positive bias. As in the case of Madrid, fines for non-compliance with the
LEZ restrictions were not planned to start before April. Therefore the effect of the LEZ is expected to be progressive, which is
consistent with the absence of bias in the beginning of the period. In addition, the 2020$^{th}$ edition of the World Mobile Congress
(the largest annual event in Barcelona, with 109.000 visitors in 2019) that takes place every year by the end of February was
officially canceled by the organizers due to the risks posed by the emerging COVID-19 pandemic. We therefore hypothesize
this cancellation contributed to the reduction of $NO_2$ levels in the city and to the slight positive bias of the ML prediction before
the lockdown.

After the lockdown, $NO_2$ mixing ratios decreased down to 8 and 11 ppbv on average at the urban background and traffic
stations, respectively, both reaching minimum daily mixing ratios of 4 ppbv. Results highlight strong and statistically signif-
icant differences with the business-as-usual situation in which $NO_2$ levels would have remained around 15-21 ppbv during
that period. As in Madrid, the strongest differences are found in April, during the phases II and III of the lockdown. Note that
these differences exceed by large the aforementioned positive bias encountered after February. Interestingly, besides the strong
reduction, observed $NO_2$ mixing ratios followed a very similar variability than business-as-usual $NO_2$, which highlights the
major influence of meteorological conditions on the levels of pollution, as previously mentioned by Tobías et al. (2020). For
instance, the increase of $NO_2$ mixing ratios between April 6$^{th}$ and April 9$^{th}$ appears linked to unusually low wind speeds over
Barcelona, 1.7 m.s$^{-1}$ on average over these days, which is slightly below the climatological (2013-2020) 5$^{th}$ percentile of wind
speed in April (1.8 m.s$^{-1}$). Without the lockdown, this stagnant situation associated with the business-as-usual emission forc-
ing would have increased $NO_2$ by about 5-10 ppbv, according to the ML predictions. Observed $NO_2$ also slightly increased
during the episode of stagnant meteorological conditions, but due to the lockdown, $NO_2$ remained at very low levels. This
event illustrates the usefulness of considering a ML model fed by meteorological data for quantifying the baseline air pollution
during the lockdown.

Over the entire lockdown period, $NO_2$ in Barcelona decreased by -6[-12,-1] ppbv (-43[-78,-5]%) at the urban background
station, regardless of the meteorological conditions. As in Madrid, a stronger reduction is found at the traffic station, with



-14[-19,-8] ppbv (-57[-80,-32]%). Therefore, in relative terms, the lockdown has induced a relatively similar decrease of NO$_2$ in both Madrid and Barcelona.

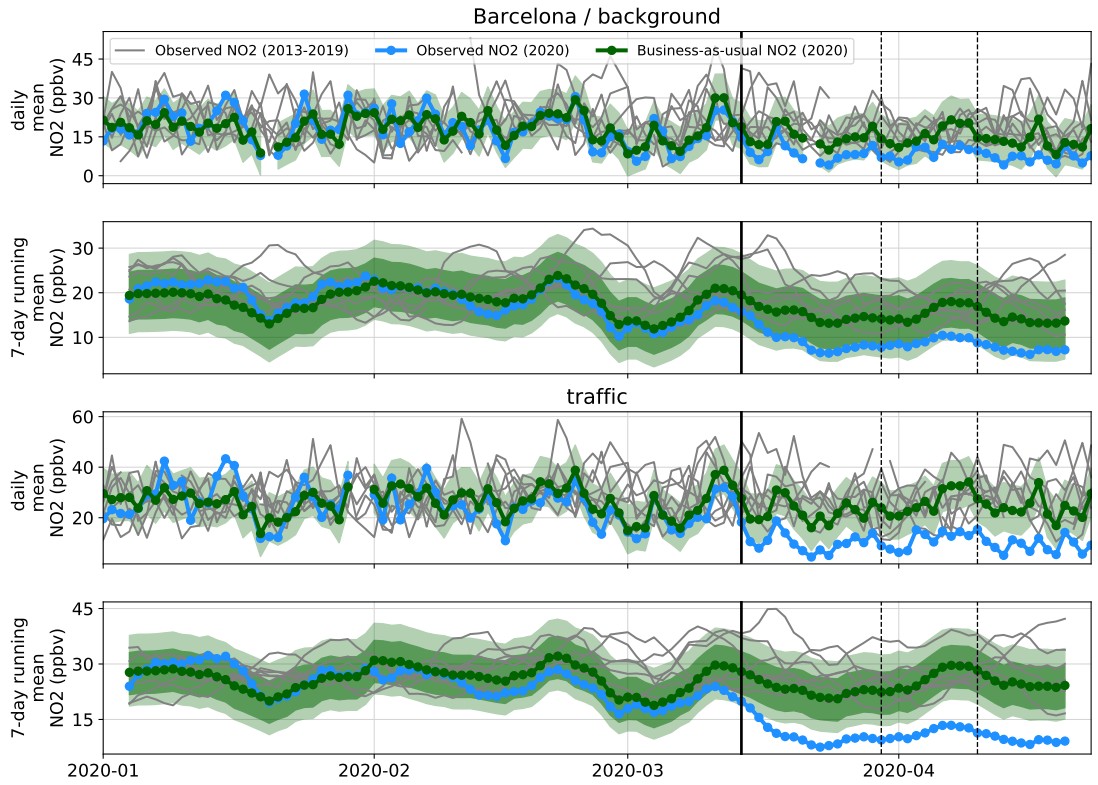

**Figure 4.** Similar to Fig. 3 in Barcelona province.

### 3.3 Meteorology-normalized changes of NO$_2$ mixing ratios over Spain

We computed the meteorology-normalized changes of NO$_2$ for all the selected stations. Results are presented in Fig. 5, together with the weekly uncertainty of our ML predictions (colored lines). For information purposes, we also display the daily uncertainty (black lines). Results are colored as a function of their degree of significance, here computed as the distance between the NO$_2$ change best estimate and the upper limit of the weekly uncertainty interval, normalized by the distance between the best estimate and zero. A degree of significance of 1 thus indicates a NO$_2$ change significant at a 90% confidence level. Statistics over the changes of NO$_2$ obtained in all provinces are reported in Table 2. A map of best estimates of NO$_2$ changes at each station is also given in Fig. 6.

Results highlight that the reduction previously described in Madrid and Barcelona extends to most Spanish provinces, although with some inter-regional variability in the extent of the change and the degree of statistical significance. On average over all





urban background stations during the entire lockdown period, $NO_2$ has decreased by -4[-8,+0] ppbv (-50[-94,+3]% in relative terms), independently from the meteorological conditions. The 5th and 95th percentiles (computed based on the mean $NO_2$

changes obtained in all provinces) are -7 ppbv (-65%) and -2 ppbv (-36%). The $NO_2$ change is significant with more than 90% confidence in 20 out of 38 provinces, with many of the remaining ones being relatively close to that confidence level. A similar, yet more statistically significant reduction is found at traffic stations, with a mean $NO_2$ decrease of -6[-11,-1] ppbv (or -50[-88,-8]%), and 28 out of 37 stations exceeding the 90% confidence level. The spread of $NO_2$ change between the different provinces is also quite similar between the two types of stations, with 5th and 95th percentiles of -66 and -28%, respectively.

Generally, the meteorology-normalized $NO_2$ reductions in the provinces of the southern half of the country appear stronger and in more cases statistically significant.

As previously observed in Madrid and Barcelona, results in Table 2 highlight noticeable differences between the different phases of the lockdown. The corresponding figures (with both absolute and relative changes) can be found in the Appendix (Figs. A1, A2, A3 and A4). The mean reduction of $NO_2$ during phase I was about -40% at both station types, and further

increased to about -55% during phases II and III. The lower reduction during the first phase is partly explained by the fact that $NO_2$ concentrations started at their business-as-usual levels and took a few days to reach their minimum. During the two last phases, $NO_2$ was found to be reduced in many more provinces, as shown by the 95th percentile that ranges between -33 and -45% depending on the type of station during phases II and III, compared to only -12 to -19% during phase I.

### 3.4 Relationship to emission reductions

We contrasted our results with a detailed $NO_x$ anthropogenic emission inventory at 4km x 4km resolution over Spain available through the bottom-up module of the HERMESv3 emission model, developed at the Earth Sciences Department of the Barcelona Supercomputing Center (Guevara et al., 2020). Averaged over the different stations considered in this study, road transport emissions are the dominant source, with 66 and 69% of the total $NO_x$ emissions in the vicinity of urban background and traffic stations, respectively. The other emission sources are the residential/commercial combustion sector (14 and 15%),

industrial point sources (8 and 13%) and shipping and port activities (11 and 3%). In Spain, the public agency in charge of monitoring traffic (*Dirección General del Tráfico*) reported progressive reductions in total traffic down to levels about -60 to -90% lower than usual, with substantial day-to-day variability and strongest reductions during weekends. Assuming to first order a linear relationship between $NO_2$ urban background mixing ratios and local surrounding $NO_x$ emissions (within a 4km x 4km cell) and applying a 70% (80%) reduction of road transport would lead to a $NO_2$ reduction of about 47% (54%), which

is consistent with our findings. Our knowledge about the impact of the lockdown on the other emission sectors remains at this stage quite limited. $NO_x$ emissions from industry likely also decreased but quantifying this reduction, even roughly, is more complex as some industries were considered as essential and thus not affected by the lockdown. Although 9-13% of the surrounding emissions (in the 4km x 4km cell of the inventory) are associated to this sector, the impact of idling industrial activities on the pollution levels observed at the selected stations may be relatively small considering that none of these stations

are classified as "industrial". The residential/commercial emission sector represents another unknown since the expected emission increment caused by a population spending more time at home may be compensated by the closure of most shops, schools



**Table 2.** Meteorology-normalized changes of NO$_2$ mixing ratios in Spain during the lockdown (phase I : 2020/03/14-2020/03/29, phase II : 2020/03/30-2020/04/09, phase III : 2020/04/10-2020/04/24). Statistics are computed based on the mean NO$_2$ changes in the different Spanish provinces.

| Change | Metric | Phases I+II+III | | Phase I | | Phase II | | Phase III | |
|---|---|---|---|---|---|---|---|---|---|
| | | Background | Traffic | Background | Traffic | Background | Traffic | Background | Traffic |
| absolute (ppbv) | mean | -4.1 | -6.3 | -3.2 | -5.4 | -5.2 | -7.3 | -4.3 | -6.8 |
| | | [-7.7,0.1] | [-10.8,-1.4] | [-6.8,1.0] | [-9.8,-0.5] | [-8.7,-1.0] | [-11.8,-2.3] | [-7.9,-0.1] | [-11.1,-1.9] |
| | std | 1.7 | 3.0 | 1.7 | 3.1 | 2.1 | 3.2 | 2.0 | 3.1 |
| | min | -9.6 | -14.2 | -8.0 | -14.0 | -11.1 | -15.3 | -10.3 | -14.8 |
| | p05 | -6.7 | -12.0 | -6.1 | -11.0 | -8.9 | -13.3 | -8.0 | -12.8 |
| | p10 | -6.3 | -10.7 | -5.3 | -9.8 | -7.3 | -11.4 | -6.9 | -11.5 |
| | p25 | -5.3 | -7.2 | -4.4 | -6.4 | -6.6 | -8.5 | -5.3 | -9.0 |
| | p50 | -3.6 | -5.6 | -2.8 | -5.0 | -4.8 | -7.0 | -4.1 | -6.0 |
| | p75 | -3.0 | -4.2 | -2.2 | -3.8 | -4.0 | -5.1 | -2.8 | -4.7 |
| | p90 | -2.2 | -3.4 | -1.5 | -1.9 | -3.0 | -3.8 | -2.0 | -3.9 |
| | p95 | -2.1 | -2.9 | -1.1 | -1.0 | -2.8 | -2.8 | -1.7 | -2.8 |
| | max | -1.1 | -1.3 | 0.6 | 0.5 | -1.1 | -1.3 | -1.4 | -2.5 |
| relative (%) | mean | -50 | -50 | -40 | -42 | -57 | -53 | -54 | -56 |
| | | [-94,3] | [-88,-8] | [-87,17] | [-80,-0] | [-96,-8] | [-89,-14] | [-100,2] | [-95,-13] |
| | std | 11 | 11 | 15 | 16 | 9 | 11 | 14 | 9 |
| | min | -68 | -71 | -64 | -69 | -72 | -71 | -77 | -74 |
| | p05 | -65 | -66 | -58 | -62 | -71 | -71 | -73 | -70 |
| | p10 | -63 | -61 | -58 | -59 | -69 | -67 | -70 | -68 |
| | p25 | -58 | -57 | -50 | -52 | -66 | -59 | -65 | -64 |
| | p50 | -50 | -50 | -41 | -43 | -55 | -56 | -55 | -55 |
| | p75 | -41 | -43 | -31 | -38 | -49 | -48 | -44 | -51 |
| | p90 | -38 | -37 | -23 | -19 | -45 | -39 | -38 | -47 |
| | p95 | -36 | -28 | -19 | -12 | -43 | -33 | -33 | -45 |
| | max | -23 | -23 | 10 | 4 | -40 | -23 | -18 | -25 |



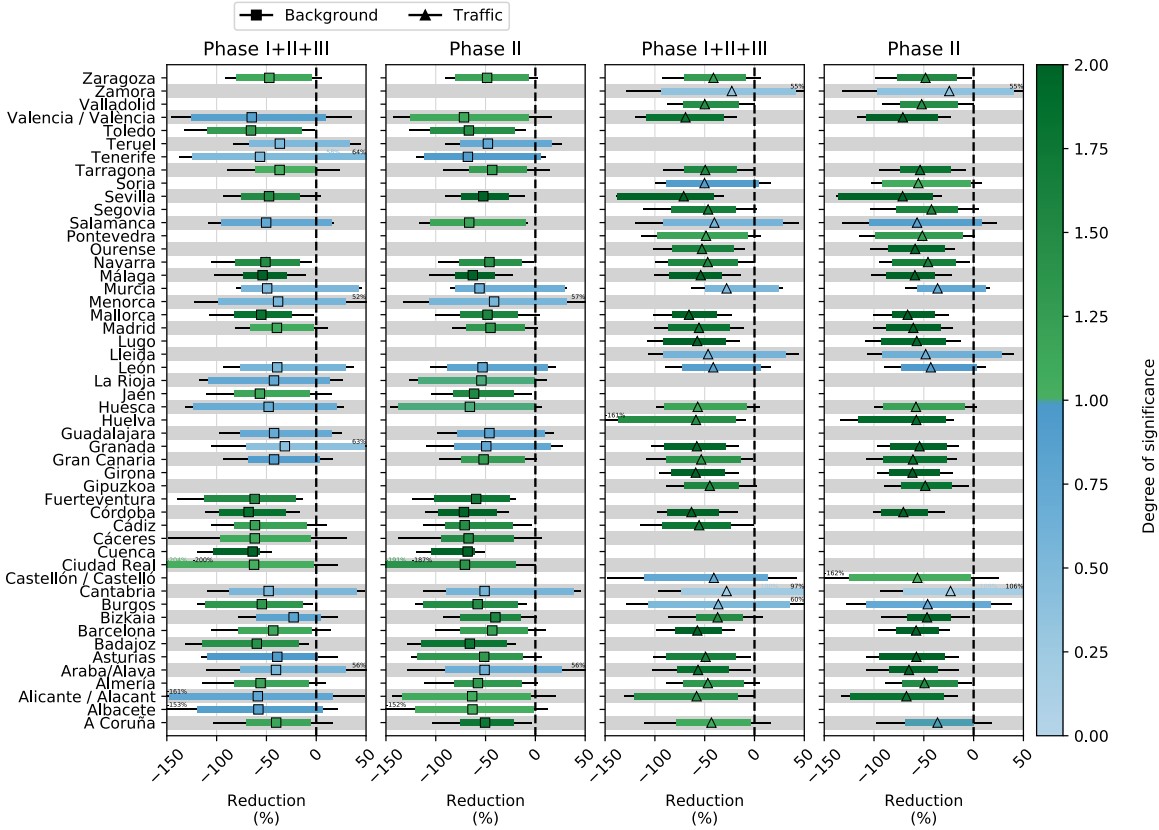

**Figure 5.** Meteorology-normalized mean $NO_2$ changes at urban background and traffic stations during the COVID-19 lockdown. Changes are shown during the entire lockdown period and during the second and most stringent phase. Best estimates and weekly uncertainties are colored according to the degree of significance (a value of 1 indicates a change statistically significant at a 90% confidence level, see text for more details). For information purposes, daily uncertainties are also indicated (in black).

and offices. A more detailed analysis of the activity data in these different emission sectors is required to better quantify how the emission forcing has been modified by the lockdown (Guevara, in preparation) and to understand the reductions of $NO_2$ obtained in this study.

Concerning traffic stations, although HERMESv3 gives a quite similar contribution of the different emission sectors compared to urban background stations, a larger contribution of road transport emissions is evidently expected since measurement instruments are deployed under the direct influence of vehicles. As a consequence, assuming that road transport is the emission sector most impacted by the lockdown (together with air traffic, but this last sector does not emit strong amounts of $NO_x$ around our set of stations), we could expect a stronger relative reduction of $NO_2$ at traffic stations, compared to urban background stations.

At first glance, Table 2 does not highlight such a difference between the two types of stations. This seems to be due to the fact that we here gather urban background and traffic stations not always collocated in the same cities, and/or located in cities of



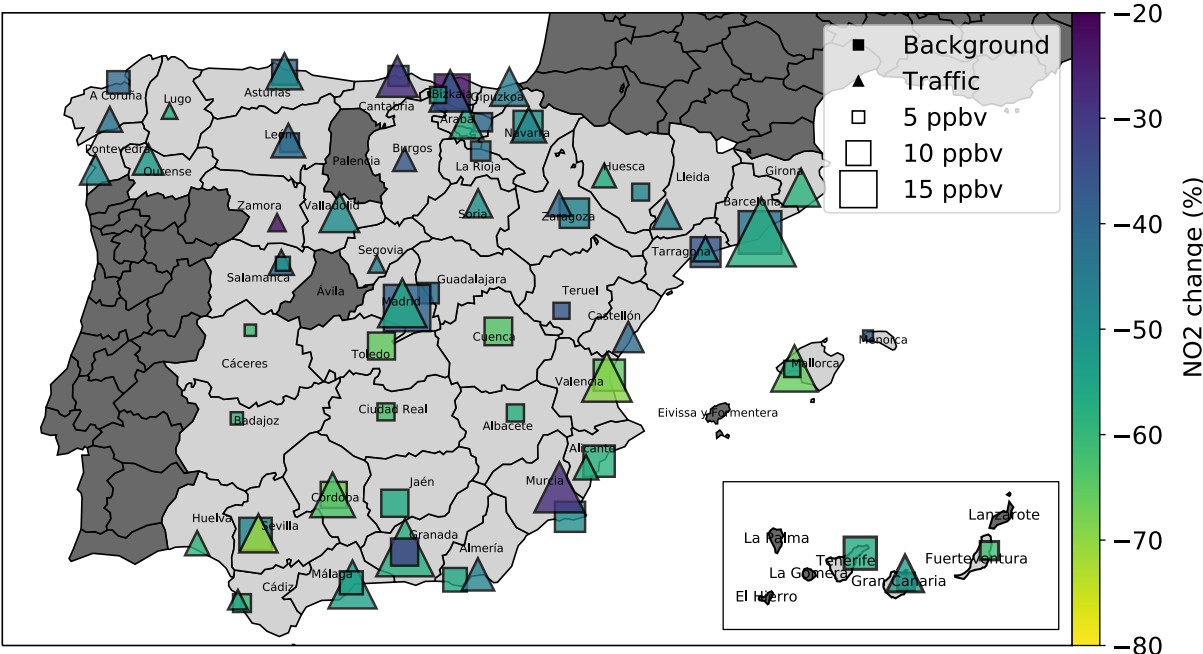

**Figure 6.** Meteorology-normalized mean NO$_2$ changes at selected urban background and traffic stations during the COVID-19 lockdown in Spain. The size of symbols is proportional to the annual average NO$_2$ mixing ratio (over 2013-2020).

very different sizes. In both Madrid and Barcelona provinces, the two selected stations are located in the same agglomeration, and results do highlight substantial differences of NO$_2$ reductions (Sect. 3.2). In total, urban background and traffic stations are collocated in the same agglomeration in 15 provinces. On average over this set of provinces, the NO$_2$ reduction is -49 and

-54% at the two types of stations, thus showing a noticeable but still relatively small difference. Focusing on the 6 largest cities within this group of provinces (Madrid, Barcelona, Valencia, Sevilla, Málaga and Mallorca), the difference of NO$_2$ reductions increases, with -51 and -62% at urban background and traffic stations, respectively. Focusing on the 2 largest cities, namely Madrid and Barcelona, the discrepancy further increases, with the NO$_2$ reductions of -41 and -57%, respectively. Therefore, results suggest that the lockdown has impacted more strongly the business-as-usual NO$_2$ levels at traffic stations than at urban

background ones, and that this difference tends to be stronger in the largest cities.

### 3.5  ML-based business-as-usual NO$_2$ versus climatological average NO$_2$

We developed the ML-based approach arguing that it allows avoiding a potentially erroneous assessment of the lockdown-related NO$_2$ changes caused by the variability of meteorological conditions. In this section, we illustrate quantitatively the benefits of our method. Besides the business-as-usual NO$_2$ daily concentrations obtained with our ML-based approach, we

consider here the mean NO$_2$ concentrations observed in 2017-2019 at this period of the year (this approach being hereafter





referred to as the climatological average approach). We compared the mean $NO_2$ concentrations obtained in each province with both approaches during the different phases of the lockdown. Taking the ML-based approach as the reference, we computed the bias of the climatological average approach. In this frame, in a given province, a small bias between the two approaches should indicate that the meteorological conditions prevailing during a given phase of the lockdown are relatively close to their

climatological values at this time of the year. For convenience, both urban background and traffic stations are gathered in this analysis.

Considering the entire lockdown period, the mean business-as-usual $NO_2$ mixing ratios predicted by the ML models averaged over all provinces is 10.3 ppbv, in close agreement with the corresponding climatological mean $NO_2$ that is 10.6 ppbv. This corresponds to a mean bias (of the climatological average approach) of only +0.3 ppbv (or +2% in relative terms). This shows

that under a business-as-usual scenario, the $NO_2$ concentrations during the lockdown period should have been close to the values typically observed at this time of the year. However, this holds at a relatively large temporal (the entire lockdown period in this case, i.e. 41 days) and spatial (all Spanish provinces) scale. Among the different provinces, the relative bias of the climatological approach ranges between -41 and +33%, with 5th and 95th percentiles of -22 and +27%, thus greatly larger than its average of +2%. This highlights the presence of substantial departures from the climatology at the province scale.

For instance, in Barcelona province, the ML-based business-as-usual and climatological mean $NO_2$ mixing ratios during the lockdown period are 15 and 19 ppbv, respectively, which corresponds to a climatological approach positively biased by +30%. Such a result is not surprising since encountering climatological conditions simultaneously in all Spanish provinces is very unlikely.

Higher when considered at the province scale, the bias of the climatological approach can also further increase when computed

over shorter time periods. Indeed, during the 3 phases of the lockdown, it gets to +12, +1.2 and +2.5%, respectively, when averaged over all provinces. Among the different provinces, the corresponding 5th/95th percentiles reach -17/+48, -29/+34 and -30/+37% during phases I, II and III, respectively. For the case of Barcelona province, these relative biases are +39, +25 and 30%.

This analysis demonstrates the need to take into account (with ML or other techniques) the meteorological variability to

accurately estimate the baseline pollution and assess the changes of pollution induced by an altered emission forcing, which appears all the more crucial when pollution changes are investigated at a fine temporal and/or spatial scale.

## 4 Conclusions

The fast spread of the COVID-19 coronavirus pushed Spanish authorities to implement a severe lockdown of the population, with drastic restrictions of social and economic activities starting on March 14th 2020. Such a situation had an impact on the

anthropogenic emissions from numerous activity sectors, some of them unambiguously (road transport and air traffic, and to a lesser extent the industrial sector), others with still unclear response (residential/commercial sector). Concomitantly, a reduction of $NO_2$ mixing ratios was reported in many locations, based on in-situ $NO_2$ measurements operated by air quality monitoring stations or space-based remote sensing (e.g. TROPOMI). Part of the reduction of $NO_2$ pollution is likely explained



by the modified emission forcing caused by the lockdown. However, the potential confounding impact of the meteorological variability (a major driver of the $NO_2$ variability) prevents to directly relate the reduction of $NO_2$ mixing ratios to the lockdown-related reduction of emissions.

To tackle this issue, we used ML models fed by meteorological data to estimate the $NO_2$ mixing ratios that would have been normally observed during the COVID-19 lockdown period under a business-as-usual emission forcing and meteorological conditions prevailing during that period. We also estimated (conservative) uncertainties affecting our ML predictions. This allowed us to quantify the changes of $NO_2$ during the lockdown that are not directly related to the variability of meteorological conditions. On average over Spain, $NO_2$ mixing ratios at urban background and traffic stations were found to decrease by about -50% due to the lockdown, with stronger reductions in phases II and III (about -55%) than in phase I (about -40%).

Due to the peculiarities of $NO_2$ (e.g. primary pollutant, short chemical lifetime, simple chemistry), we expect these changes to be mainly driven by the reduction of $NO_x$ anthropogenic emissions. Considering that the lockdown also impacted the emissions of numerous other chemical compounds, an alteration of the business-as-usual chemical fate of $NO_2$ (through a modification of its oxidation into nitric acid) cannot be excluded. However, we are considering here urban stations located close to the $NO_x$ emission sources, where this effect is likely small compared to the reduction of direct emissions.

Regarding our methodology, we note that the COVID-19 lockdown and the associated changes of pollutants like particulate matter should have also altered the meteorological conditions by perturbing the radiative fluxes and clouds. Indeed, this methodology precludes the remote and local influences of lockdown-related air pollution changes upon local weather. In any case, given the chaotic nature of the atmosphere and the long duration of the lockdown, it would be indeed impossible to know the weather conditions that would had been observed during the lockdown in a business-as-usual scenario.

It is also worth noting that the quality of the ERA5 meteorological data may have deteriorated due to the lockdown through the strong reduction of air traffic. Indeed, although satellites remain the dominant provider of meteorological observations, commercial aircraft provide valuable amounts of in-situ meteorological observations in the troposphere and lower stratosphere, especially for wind speed. However, some meteorological services are currently operating additional atmospheric soundings to compensate this loss of data. In any case, the impact on the meteorological conditions close to the surface is probably limited. In this work, we analyzed the $NO_2$ data available in Spain over the first 41 days of lockdown, which includes the phase of most stringent lockdown in early April. At the date of submission of this study, the lockdown was still on-going in Spain, with restrictions planned to be progressively relaxed until late June at least. Indeed, the impact of the lockdown upon air pollution levels will likely extend way beyond the period considered in this study. Besides the direct effects of the lockdown-related restrictions, the foreseen economic downturn whose size, length and characteristics are still uncertain may also substantially affect the levels of $NO_2$ pollution, as already observed following the 2008-2009 economic recession, with one-year recession-driven $NO_2$ reductions of 10-30% across Spain and Europe (Castellanos and Boersma, 2012).

In a separate study, our meteorology-normalized estimates are used to quantify the circumstantial reduction in the mortality attributable to the short-term effects of $NO_2$ during the lockdown (Achebak et al., submitted).



*Code and data availability.* The EEA AQ eReporting, ERA5 and Gridded Population of the World (GPW) version 5 datasets used in this study are publicly available. The HERMESv3_BU (Bottom-Up) code package with its documentation is publicly available at the following gitlab repository: https://earth.bsc.es/gitlab/es/hermesv3_bu (https://doi.org/10.5281/zenodo.3521897, Guevara et al., 2019).

## Appendix A: Quality Assurance (QA) applied to NO$_2$ dataset


Using the information provided by GHOST (Globally Harmonised Observational Surface Treatment; Bowdalo, in preparation), we applied numerous QA screening to the NO$_2$ dataset, in order to remove : missing measurements (flag 0), infinite values (flag 1), negative measurements (flag 2), zero measurements (flag 4), measurements associated with data quality flags given by the data provider which have been decreed by the GHOST project architects to suggest the measurements are associated with substantial uncertainty or bias (flag 6), measurements for which no valid data remains to average in temporal window after screening by key QA flags (flag 8), measurements showing persistently recurring values (rolling 7 out of 9 data points; flag 10), concentrations greater than a scientifically feasible limit (above 5000 ppbv) (flag 12), measurements detected as distributional outliers using adjusted boxplot analysis (flag 13), measurements manually flagged as too extreme (flag 14), data with too coarse reported measurement resolution (above 1.0 ppbv) (flag 17), data with too coarse empirically derived measurement resolution (above 1.0 ppbv) (flag 18), measurements below the reported lower limit of detection (flag 22), measurements above the reported upper limit of detection (flag 25), measurements with inappropriate primary sampling for preparing NO$_2$ for subsequent measurement (flag 40), measurements with inappropriate sample preparation for preparing NO$_2$ for subsequent measurement (flag 41) and measurements with erroneous measurement methodology (flag 42).

## Appendix B: Decision tree-based ensemble methods


Among the myriad of ML models available nowadays, we opted for decision tree-based ensemble methods. The general idea of ensemble methods is to combine an ensemble of independent base learners (or weak learners). Base learners here designate simple models that perform only slightly better than a random guessing. Decision trees are currently the base learner most commonly used in ML ensemble methods (but other types of learners could be possible). Given a training dataset and a regression problem, one characteristic of decision trees lies in the fact that it is always possible to reach a high accuracy (by growing a large enough tree) but at the cost of very poor generalization skills. In ML terminology, such large trees are said to have a small bias but a large variance. To be appropriate base learners, decision trees used in ensemble methods are thus constrained to have a low number of branches (sometimes referred to as trunks), which increases the bias but reduces the variance. The strength of ensemble methods then stems out from the fact that combining a sufficiently large number of base learners (of quite poor performance individually) allows to reach an enhanced performance in addition to better generalization skills, the corresponding ensemble being less unstable to the addition of new data.

Once the form of the base learner is chosen, a strategy is required for building this ensemble of *independent* base learners. Three main approaches have been proposed over the past: (i) bagging, (ii) boosting, (iii) random forests (RF). Bagging consists in aggregating base learners trained on a bootstrap sample of the training dataset. Boosting consists in aggregating base learners





trained on different labels: the first base learner is trained on the dataset, the second on the errors left by the previous one, the
third on the errors left by the two previous ones, and so on. RF (used by Grange et al. (2018) and Grange and Carslaw (2019))
consists in aggregating base learners trained on random subsets of the training dataset based on a random subset of features.

**Appendix C:  Tuning of the GBM model**

The training of the model is conducted together with a search of the optimal hyperparameter tuning. We retained a so-called
*randomized search* in which a range of values is given for each hyperparameter of interest and a total number of hyperparame-
ters combinations to test (20 in our case). Compared to the so-called *grid search* in which all combinations of hyperparameters
are tested, this choice allows to explore a large part of the hyperparameters space for a greatly reduced computational cost, and
is less prone to overfitting.

We used the *scikit-learn* Python package. The learning rate was fixed to 0.05 and the number of features to consider when look-
ing for the best split is fixed to the square root of the number of features (*max_features* in *scikit-learn*, set to "sqrt"). Besides
that, the tuning of the GBM model was done over the following set of hyperparameters: the tree maximum depth (*max_depth*
in the *scikit-learn* Python package: values from 1 to 5 by 1), the subsample (*subsample* : from 0.3 to 1.0 by 0.1), the number of
trees (*n_estimators*: from 50 to 1000 by 50) and the minimum sample in terminal leaves (*min_samples_leaf* : from 1 to 30). The
maximum depth (or the maximum number of subsequent splits in the individual decision trees) controls how much interaction
between the features can be taken into account. The subsample hyperparameter represents the fraction of samples to be used for
fitting an individual base learner. Values below unity correspond to the so-called *stochastic gradient boosting* and usually allow
to decrease the variance at the cost of an increased bias (low values also allow to speed up the training phase). The minimum
sample leaf hyperparameter controls the minimum number of samples to allow in a terminal node (larger values limiting the
risk of overfitting).





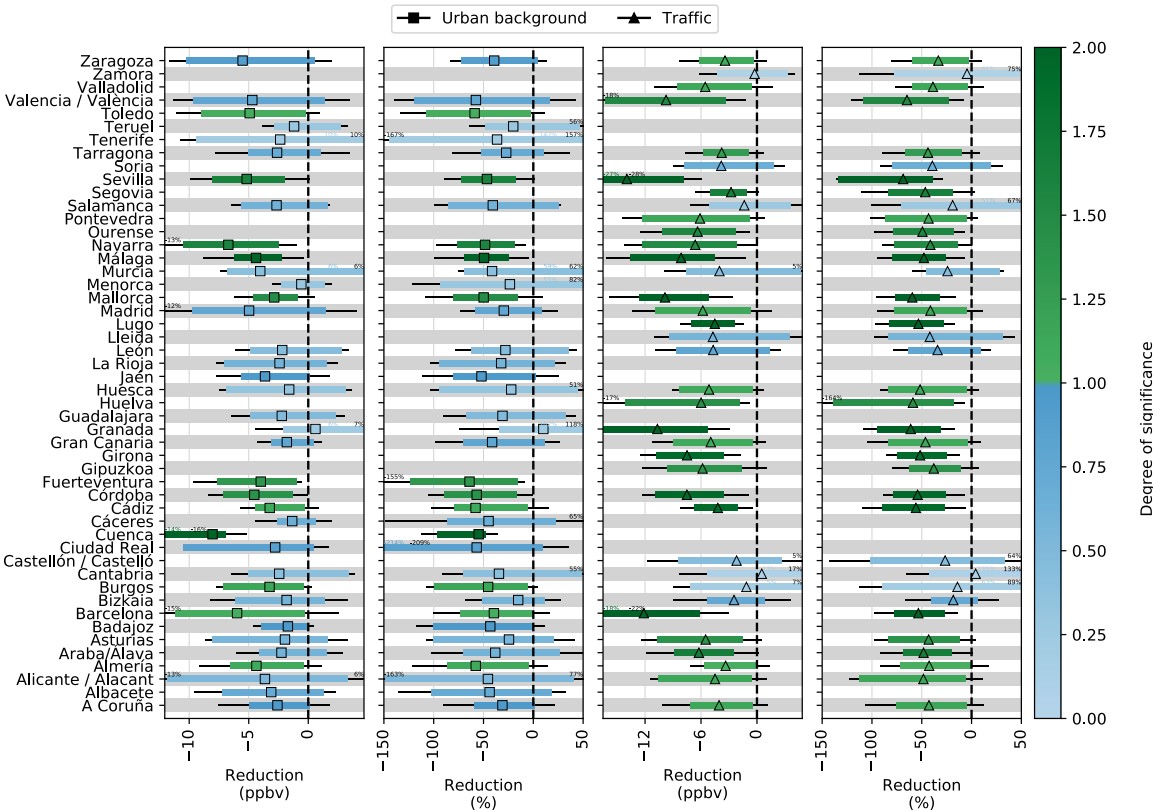

**Figure A1.** Absolute and relative meteorology-normalized NO$_2$ changes during phase I of the lockdown (2020/03/14-2020/03/29), at urban background (left panels) and traffic stations (right panels). The uncertainties shown with colored bars correspond here to the 90% confidence level interval computed at the weekly scale. For information purposes, the uncertainties affecting the ML-based daily predictions are also shown (black bars).





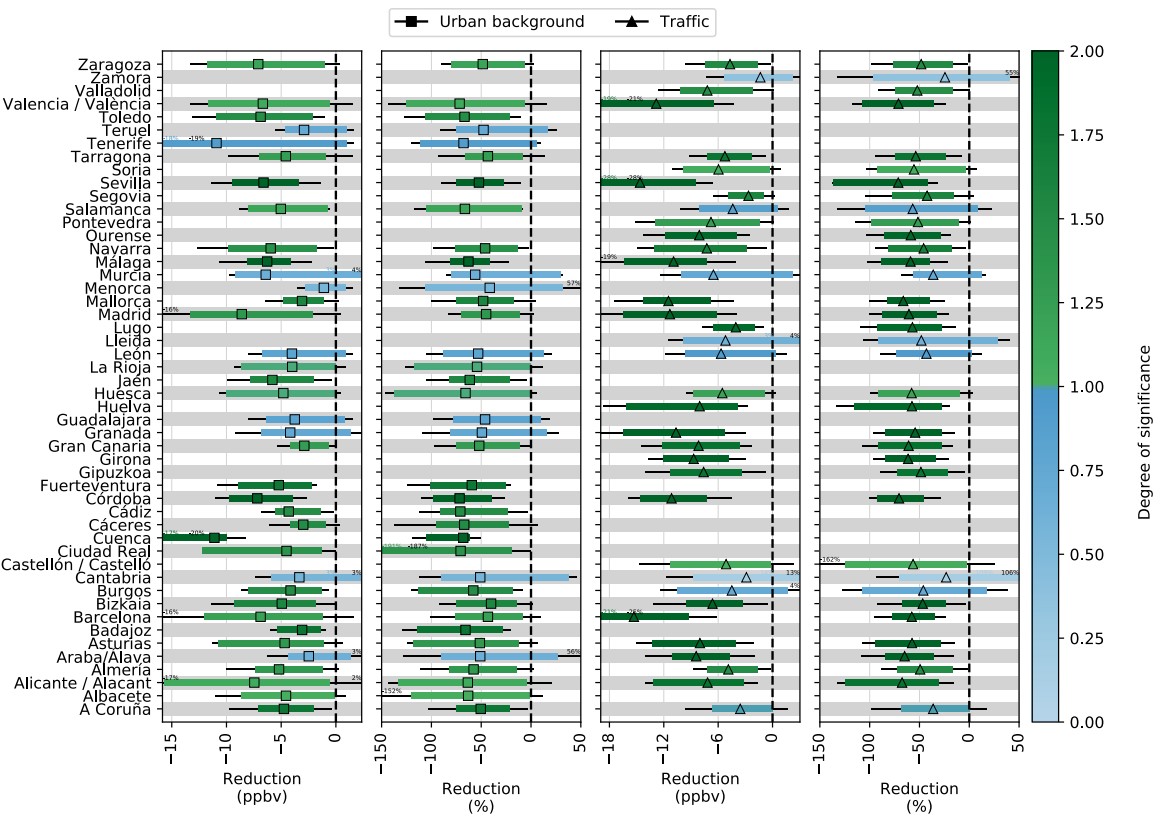

**Figure A2.** Similar to Fig. A1 for the phase II of the lockdown (2020/03/30-2020/04/09).





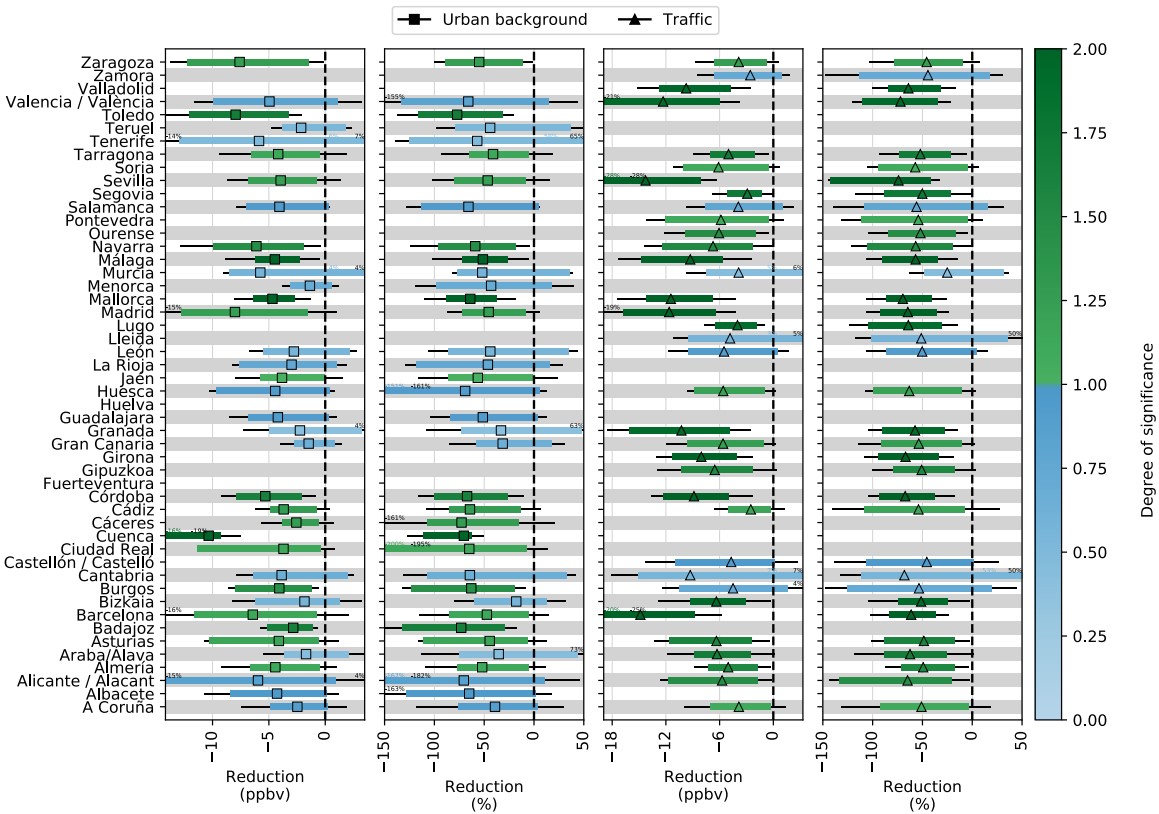

**Figure A3.** Similar to Fig. A1 for the phase III of the lockdown (2020/04/10-2020/04/23).





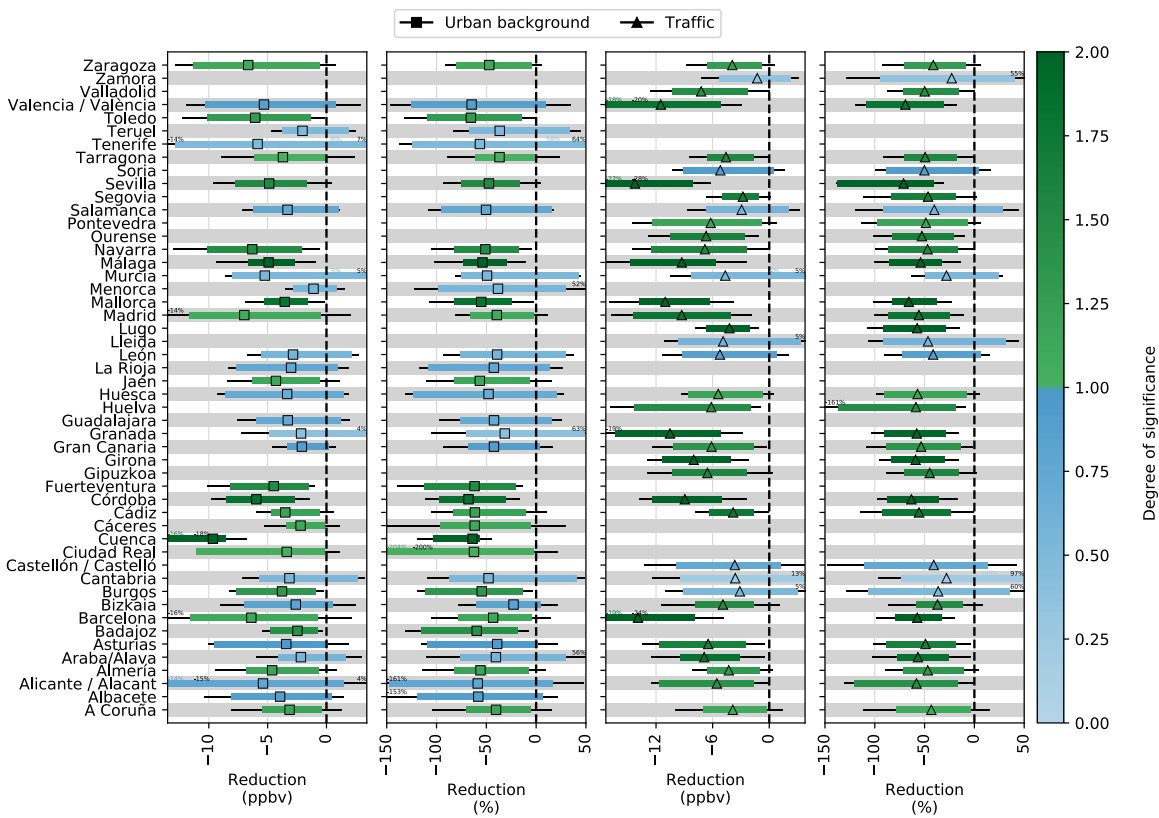

**Figure A4.** Similar to Fig. A1 for the entire lockdown period (2020/04/13-2020/04/23).





**Table A1.** Stations selected in each Spanish province.

| Province | Urban background station | Traffic station |
|---|---|---|
| A Coruña | ES1957A Torre De Hércules (43.382800, -8.409200) | ES1901A San Caetano (42.887800, -8.531100) |
| Albacete | ES1535A Albacete (38.979300, -1.852100) | - |
| Alicante / Alacant | ES1915A Alacant-Florida-Babel (38.340278, -0.506667) | ES1849A Elx-Parc De Bombers (38.259167, -0.717500) |
| Almería | ES1549A El Ejido (36.769720, -2.810970) | ES1393A Mediterráneo (36.841330, -2.446720) |
| Araba/Álava | ES1544A Agurain (42.849000, -2.393700) | ES1492A Tres Marzo (42.856070, -2.667790) |
| Asturias | ES1974A Montevil (43.516600, -5.670700) | ES1272A Constitución (43.529900, -5.673500) |
| Badajoz | ES1819A Merida (38.907500, -6.338060) | - |
| Barcelona | ES1396A Barcelona (Sants) (41.378803, 2.133098) | ES1438A Barcelona (L'Eixample) (41.385343, 2.153822) |
| Bizkaia | ES1713A Parque Europa (43.254900, -2.902300) | ES1244A Mazarredo (43.267500, -2.935200) |
| Burgos | ES1598A Zalla (43.212910, -3.134400) | ES1160A Burgos 1 (42.350830, -3.675560) |
| Cantabria | ES1529A Tetuán (43.467780, -3.790280) | ES1580A Santander Centro (43.460560, -3.808610) |
| Castellón / Castelló | - | ES1834A Castelló-Patronat D'Esports (39.988889, -0.026111) |
| Ciudad Real | ES1857A Ciudad Real (38.993900, -3.937800) | - |
| Cuenca | ES1858A Cuenca (40.061900, -2.129700) | - |
| Cáceres | ES1997A Plasencia (40.077780, -6.147220) | - |
| Cádiz | ES1593A San Fernando (36.460590, -6.203070) | ES1479A Avda. Marconi (36.506020, -6.268570) |
| Córdoba | ES1799A Lepanto (37.892610, -4.762340) | ES2047A Avda. Al-Nasir (37.892600, -4.780100) |
| Fuerteventura | ES1978A Casa Palacio-Puerto Del Rosario (28.498380, -13.860830) | - |
| Gipuzkoa | - | ES1494A Ategorrieta (43.322000, -1.960700) |
| Girona | - | ES1999A Girona (Escola De Música) (41.976386, 2.816547) |
| Gran Canaria | ES1919A Parque De San Juan-Telde (28.003645, -15.411851) | ES1573A Mercado Central (28.133732, -15.432823) |
| Granada | ES1973A Ciudad Deportiva (37.135560, -3.619250) | ES1560A Granada - Norte (37.196100, -3.612660) |
| Guadalajara | ES1536A Azuqueca De Henares (40.571000, -3.264600) | - |
| Huelva | - | ES1340A Pozo Dulce (37.253360, -6.935140) |
| Huesca | ES2041A Monzón Centro (41.916140, 0.191101) | ES1417A Huesca (42.136110, -0.403890) |
| Jaén | ES1656A Ronda Del Valle (37.782550, -3.781570) | - |
| La Rioja | ES1602A La Cigüeña (42.464000, -2.428000) | - |
| León | ES1988A León 4 (42.575278, -5.566389) | ES1161A Barrio Pinilla (42.603889, -5.587222) |
| Lleida | - | ES1225A Lleida (Irurita - Pius Xii) (41.615795, 0.615726) |
| Lugo | - | ES1905A Lugo-Fingoy (42.997900, -7.550900) |
| Madrid | ES1941A Ensanche De Vallecas (40.372780, -3.611944) | ES1938A Castellana (40.439722, -3.690278) |
| Mallorca | ES1604A Bellver (39.563320, 2.620550) | ES1610A Foners (39.570080, 2.655830) |
| Menorca | ES1828A Ciutadella De Menorca (40.009440, 3.856480) | - |
| Murcia | ES1921A Mompean (37.603056, -0.975278) | ES1633A San Basilio (37.993611, -1.144722) |
| Málaga | ES1751A El Atabal (36.729560, -4.465530) | ES2031A Avenida Juan Xxiii (36.707300, -4.446000) |
| Navarra | ES1472A Iturrama (42.807220, -1.651390) | ES1740A Plaza De La Cruz (42.812220, -1.640000) |
| Ourense | - | ES1096A Gomez Franqueira (42.353000, -7.877900) |
| Pontevedra | - | ES1137A Arenal (42.219000, -8.742100) |
| Salamanca | ES1889A Salamanca 6 (40.960833, -5.639722) | ES1618A Salamanca 5 (40.979167, -5.665278) |
| Segovia | - | ES1967A Segovia 2 (40.955556, -4.110556) |
| Sevilla | ES1425A Principes (37.375250, -6.005580) | ES0817A La Ranilla (37.384250, -5.959620) |
| Soria | - | ES1643A Soria (41.766667, -2.466667) |
| Tarragona | ES1666A Tarragona (Parc De La Ciutat) (41.117388, 1.241650) | ES1124A Tarragona (Sant Salvador) (41.159450, 1.239704) |
| Tenerife | ES1975A Depósito Tristán-Sta Cruz De Tf (28.458160, -16.278776) | - |
| Teruel | ES1421A Teruel (40.336390, -1.106670) | - |
| Toledo | ES1818A Toledo2 (39.868100, -4.020800) | - |
| Valencia / València | ES1885A València-Politècnic (39.480300, -0.336400) | ES1239A València-Pista De Silla (39.456111, -0.375833) |
| Valladolid | - | ES1631A Arco De Ladrillo Ii (41.645556, -4.730278) |
| Zamora | - | ES1927A Zamora 2 (41.509722, -5.746389) |
| Zaragoza | ES1641A Renovales (41.635280, -0.893610) | ES1418A Alagón (41.762780, -1.143330) |



*Author contributions.* Contributed to conception and design: HP, CPG-P. Contributed to acquisition of data: DB, KS. Contributed to analysis and interpretation of data: HP, DB, CPG-P, MG, AS, OJ. Drafted the article: HP, CPG-P.

*Competing interests.* The authors declare that they have no conflict of interest.

*Acknowledgements.* This project has received funding from the European Union's Horizon 2020 research and innovation programme under the Marie Sklodowska-Curie grant agreement H2020-MSCA-COFUND-2016-754433. We also acknowledge support by the European Research Council (grant no. 773051, FRAGMENT), the AXA Research Fund, the Spanish Ministry of Science, Innovation and Universities (RYC-2015-18690, CGL2017-88911-R, RTI2018-099894-B-I00 and Red Temática ACTRIS España CGL2017-90884-REDT), the BSC-CNS "Centro de Excelencia Severo Ochoa 2015-2019" Program (SEV-2015-0493), PRACE for awarding us access to Marenostrum Supercomputer in the Barcelona Supercomputing Center, and H2020 ACTRIS IMP (871115).



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
