# Peer review of "Meteorology-normalized impact of the COVID-19 lockdown upon $NO_2$ pollution in Spain"

_Atmospheric Chemistry and Physics, 2020_

## Referee Comment (RC1) · Anonymous Referee #2 · 22 Jun 2020

This work by Petetin et al., deals with the hot topic of variation of pollutants during the lockdown measures against the COVID19 pandemic. More specifically it focuses on the NO2 and the area of the Spanish state. Transports are the main source of NO2 in the troposphere, thus the reduction of traffic is estimated to lower significantly the emissions. Though the decrease of the emissions was very clear during the lockdown, the actual concentration in various areas is also dependent on meteorological parameters that rule the dispersion and the chemical processes of the gas. In order to better estimate the expected concentrations, based on meteorology, authors have trained a machine learning algorithm, to simulate the business as usual conditions, using as input meteorological variables. The work is generally well presented and should be accepted for publication in ACP after minor revisions.

[Figure]

Specific comments

L10 It would be better to provide some quantitative measure of the performance of the model. L77 Please provide some bibliographical reference for the uncertainty of these NO2 measurements. L100 The selection of variables to feed the ML algorithm is very crucial and implies the physical and chemical processes that should be associated with the gas' concentration. My thought is that the photochemical cycle is implied by cloud coverage, which indirectly influences the irradiance which drive the photolysis. Since daily values are used, it is imperfectly fed to the algorithm, since nighttime cloud coverage would no affect NO2 concentration. Thus, some irradiance related variable from ERA-5 seems a better choice (SSI is a good one to investigate first). Since the results are satisfactory even using the cloud coverage proxy, I suggest to add some discussion on the selection of the variables and probable investigate other ones in the future. Figure 1. I think it is somehow difficult to understand the map, probably a different selection of colorbar would make it easier to figure out the conditions. L.119 ERA-5 spatial resolution is around 30km. Are there stations that correspond to the same grid point of the database? Please discuss the uncertainty introduced by the problem of non-colocation of ERA-5 and actual measuring stations. L130 Is that the case in any of the data used here? Are there any stations with significant trends in the training period? L141 Following the arguments deployed in previous paragraphs, it seems preferable to test the validity in the same period of the year, as the one of interest (March-May), than in January -February. L159 Figure 1 shows that a number of stations have mean concentrations ∼5ppvb. Thus these intervals are very huge, making the result not reliable. I suggest to present these intervals in a different way and not averaging all tha data. L167-168 This argument is note clear. Please explain in detail Table1 The test cases N seems very low, are these implying number of stations or total number of test days for all stations? L255 In some cities, such zones, resulted in much higher traffic in peripheral road networks. Thus the stations at 3 and 9 km, might experiencing heavier traffic due to LEZ in the center. This should be answered locally by explaining the main routes and the traffic of each city. L263 "Statistically significant" should not be used

without proper definition and explanation. Explain which significance tests you used, what was the outcome and then provide such conclusions. 3.3 I think it is important to present some representative cases of other stations' timeseries in figures similar to 3 and 4. These provide a very clear picture of the conditions during the lockdown phases. Are there any periods of higher than business as usual concentration, probably in the stations with low mean values (Granada and Murcia probably)? 3.5 A figure showing all three timeseries (climatological, business as usual and measured) would be very useful, at least for some representative stations. L.384-387 This is a very important finding at should be highlighted more and included in the conclusions, because it is general for future application of climatological values. L445 It is not clear if all the flagged data were removed for the process or if different flags were treated differently.

---

## Referee Comment (RC2) · Anonymous Referee #1 · 7 Jul 2020

The article under review here aims to quantify the impact of the Covid-19 lockdown measures in Spain on air quality. The topic is interesting from the point of view of air quality practitioners and the general public, but it also raises substantial scientific challenges. Even if economic activities were substantially reduced during the lock down period, the impact of meteorological factors on air quality precludes a simple comparison with previous years. Instead, the authors mobilize innovative machine learning approaches to tackle the issue.

The quality of the presentation, scientific quality, and societal relevance are excellent, and publication in ACP is therefore recommended.

I am nevertheless proposing the following minor suggestions that could help further strengthen the paper.

[Figure]

General comment:

The authors should be encouraged to extent the coverage of their study. Applying the method over the whole of Europe is certainly the scope for another paper. But an extension of the temporal coverage up to the end of the lockdown in Spain would be interesting.

Specific comments:

L24, L403: the coronavirus is SARS-COV-2 not COVID-19

L36: without supporting reference, it is wiser to state that "the impact on industry is *presumably* more contrasted"

L50: in the motivation of the work, the authors could add that this type of analysis will serve to validate the model-based assessment using emission scenarios derived from activity data during the lockdown

L69: where is the GHOST data available ? If GHOST database is not publicly open, the reference of the availability of the data should remain EEA's AQ e-reporting database.

L75: the formal deadline for 2019 AQ e-reporting data to be delivered as E1a is September 2020, what is the fraction of 2019 data already E1a at the date of submission?

L125: please clarify what you mean by "unique values", is the date index the julian day, and if so why would it be unique?

L145: hyperparameters should be defined and discussed either in the main text or in the annex. Further details would be appreciated in the annex on how the choice of those hyperparameters are related with the problem at hand (density and spread of observations, number and diversity of predictors etc. . .).

L245: include the value of the uncertainty interval, it is difficult to compare percentages in 3.2 and ppbv intervals in 2.3.3

[Figure]
* * *
**Interactive
comment**

L255: the impact of the LEZ could actually be an increase of NO2 at stations in the outskirts of that zone

Figure 2: N seems to be missing from the plot

L266: clarify if the confidence interval is taken from the distribution of daily differences

L325 and L344: could there be a role of background ozone in the relation between NOx emission changes and NO2 concentrations that would appear through this latitudinal gradient?

L365: clarify which reduction is for urban and traffic stations

L412: also mention day of the week in the preductors, which is presumably very important for NO2

---

## Author Comment (AC1) · 31 Aug 2020

Comments from the reviewer are in blue, and answers in black (text citations and modifications are highlighted in italics).

This work by Petetin et al., deals with the hot topic of variation of pollutants during the lockdown measures against the COVID19 pandemic. More specifically it focuses on the NO2 and the area of the Spanish state. Transports are the main source of NO2 in the troposphere, thus the reduction of traffic is estimated to lower significantly the emissions. Though the decrease of the emissions was very clear during the lockdown, the actual concentration in various areas is also dependent on meteorological parameters that rule the dispersion and the chemical processes of the gas. In order to better estimate the expected concentrations, based on meteorology, authors have trained a machine learning algorithm, to simulate the business as usual conditions, using as input meteorological variables. The work is generally well presented and should be accepted for publication in ACP after minor revisions.

We thank the reviewer for his/her constructive comments.

Specific comments
- L10 It would be better to provide some quantitative measure of the performance of the model. We modified the sentence : "*The ML predictive models were found to perform remarkably well in most locations, with overall bias, root-mean-squared error and correlation of +4%, 29% and 0.86.*"
- L77 Please provide some bibliographical reference for the uncertainty of these NO2 measurements. We added some information regarding the measurement uncertainties : "*All $NO_2$ measurements taken into account here are operated using chemiluminescence with an internal Molybdenum converter. Although predominantly used over Europe for measuring $NO_2$, this measurement technique is well known to be have strong positive artifacts due to interferences of $NO_z$ compounds (e.g. nitric acid, peroxyacetyl nitrates, organic nitrates), especially during daytime when these species are photo-chemically formed, up to a factor of 2-4 as observed during summertime in urban atmospheres (e.g. Dunlea et al., 2007; Villena et al., 2012). In our case, the positive artifacts at urban background stations are probably lower since the period of study (late winter and early spring) is less photo-chemically active than summertime. Even lower interferences are expected at traffic stations where the $NO_z/NO_x$ ratio is typically lower due to the proximity to fresh NOx emissions. In any case, the present study focuses on the relative changes of $NO_2$ due to the lockdown, so biases in the $NO_2$ measurements are of lower importance.*" with the corresponding references are :
    - Dunlea, E. J., Herndon, S. C., Nelson, D. D., Volkamer, R. M., San Martini, F., Sheehy, P. M., Zahniser, M. S., Shorter, J. H., Wormhoudt, J. C., Lamb, B. K., Allwine, E. J., Gaffney, J. S., Marley, N. A., Grutter, M., Marquez, C., Blanco, S., Cardenas, B., Retama, A., Ramos Villegas, C. R., Kolb, C. E., Molina, L. T., and Molina, M. J.: Evaluation of nitrogen dioxide chemiluminescence monitors in a polluted urban

environment, Atmos. Chem. Phys., 7, 2691–2704, doi:10.5194/acp-7-2691-2007, 2007.

o Villena, G., Bejan, I., Kurtenbach, R., Wiesen, P., and Kleffmann, J.: Interferences of commercial NO2 instruments in the urban atmosphere and in a smog chamber, Atmos. Meas. Tech., 5, 149–159, doi:10.5194/amt-5-149-2012, 2012.

- L100 The selection of variables to feed the ML algorithm is very crucial and implies the physical and chemical processes that should be associated with the gas' concentration. My thought is that the photochemical cycle is implied by cloud coverage, which indirectly influences the irradiance which drive the photolysis. Since daily values are used, it is imperfectly fed to the algorithm, since nighttime cloud coverage would no affect NO2 concentration. Thus, some irradiance related variable from ERA-5 seems a better choice (SSI is a good one to investigate first). Since the results are satisfactory even using the cloud coverage proxy, I suggest to add some discussion on the selection of the variables and probable investigate other ones in the future. The reviewer here raises an interesting point, and we agree that including such information is susceptible to improve the ML-based predictions. We thus re-run our analysis adding the ERA5 surface net solar radiation, surface solar radiation downwards and the downward UV radiation at the surface to the set of features. The impact on the statistical results is generally positive although relatively small (error and correlation very slightly improved, and bias very slightly increased). On average, the importance of these new features is 4, 4 and 5%, respectively, which demonstrates their usefulness for predicting NO2 concentrations. We updated the entire document (figures, tables and text) with the results obtained with this new set of features. Note that most changes are minor, so the discussion remains the same. We thank the reviewer for helping us further improving the results.

- Figure 1. I think it is somehow difficult to understand the map, probably a different selection of color bar would make it easier to figure out the conditions. The *viridis* default color bar in Python *matplotlib* library presents a number of well recognized advantages over most of the existing color bars (e.g. color-blind friendly, perceptually uniform when printed in black and white). We thus decided to keep it but we modified the number of colors in order to make Figs. 1 and 6 easier to read.

- L119 ERA-5 spatial resolution is around 30km. Are there stations that correspond to the same grid point of the database? Please discuss the uncertainty introduced by the problem of non-colocation of ERA-5 and actual measuring stations. Given the ERA5 spatial resolution, urban background and traffic stations within a same city typically belong to the same ERA5 grid cell. We are not sure to perfectly understand the point raised here by the reviewer given that ERA5, as gridded data, can always be collocated with any measuring stations. After that, considering numerical meteorological data over a volume (the grid cell) as a proxy of the meteorological conditions occurring at a point (the air quality station) indeed necessarily comes with some uncertainties. The uncertainties (e.g. of representativeness) related to the relatively coarse resolution of ERA5 for representing accurately the meteorological conditions at the different

stations are already discussed (L216-226 in the first version) in the initial manuscript, so we think that there is not much more useful information to add concerning this point.

- L130 Is that the case in any of the data used here? Are there any stations with significant trends in the training period? To our opinion, the 3-years training period is too short to compute meaningful trends. Over the period 2013-2019, a simple linear trend analysis on annual mean NO2 mixing ratios indicates that 21 over 75 stations show significant trends, with a median of -5%/year.

- L141 Following the arguments deployed in previous paragraphs, it seems preferable to test the validity in the same period of the year, as the one of interest (March-May), than in January -February. The reviewer is raising here an important point that deserves more discussion. In the revised version of the paper, we greatly reshaped Table 1 and the corresponding discussion.

As explained in the text, at each station, several ML experiments have been conducted, including the reference one with training over 2017-2019 and testing in 2020 (hereafter referred to as the $EXP_{2020}$ experiment), and the four other experiments based on past data and used for quantifying the uncertainties of our NO2 predictions (hereafter referred to as the $EXP_{2016}$, $EXP_{2017}$, $EXP_{2018}$, and $EXP_{2019}$ experiments).

Only the ML models obtained from the reference $EXP_{2020}$ experiment are used for estimating the business-as-usual NO2 during the COVID-19 lockdown, which explains why we initially focused on them for the statistical evaluation. Since the lockdown period in 2020 can evidently not be used for evaluation, this constrained us to restrict the evaluation to the period 01/01/2020-13/03/2020. However, we agree that the performance of the ML models may be different during the lockdown period. In the revised version of the paper, we now also discuss the performance obtained with the four other experiments ($EXP_{2016-2019}$), which allows to check the performance during the period of the year of the lockdown. Besides Table 1, the text in this section is modified as follows :

"*The performance of the ML predictions in each Spanish province and station type is shown in Fig. 2, and the statistics over all Spanish provinces reported in Table 1. Statistical results in Table 1 are given for both the reference ML experiment (EXP2020) and the other experiments combined together (EXP2016, EXP2017, EXP2018 and EXP2019, hereafter referred to as EXP2016–2019). Besides providing a broader view of the performance of our modeling strategy, considering these past experiments also allows assessing the performance of the ML predictions during the period of the year of the lockdown (14/03-30/04, for years 2016 to 2019), which may be important given the potential seasonality of prediction errors. Statistics obtained at urban background and traffic stations are given in Table A2 in Appendix. Results are evaluated using the following metrics, calculated based on daily NO2 mixing ratios : mean bias (MB), normalized mean bias (nMB), root mean square error (RMSE), normalized root mean square error (nRMSE) and Pearson correlation coefficient (PCC).*

*For information purposes, we included the statistical results obtained over the training dataset (2017/01/01-2019/12/31 in EXP$_{2020}$). Checking results over the training data may be useful for highlighting obvious situations of overfitting, when the performance is almost perfect. At both urban background and traffic stations, results show no bias, low nRMSE (always below 35%, 19% when considering all provinces), and a high PCC of 0.96. Similar results are obtained when considering the ensemble of all past experiments (EXP$_{2016-2019}$). Although such a performance obtained is very good, there are no clear signs of too prejudicial overfitting at this stage.*

*On the test dataset of the EXP$_{2020}$ reference experiment (2020/01/01-2020/03/13, before the lockdown), the performance remains reasonably good in most provinces. Over all Spanish provinces, the nMB increases to +4%, the nRMSE to 29% and the PCC is reduced to 0.86, in very close agreement with the performance obtained with EXP$_{2016-2020}$ (nMB of +1%, nRMSE of 28% and PCC of 0.86). In comparison, the performance obtained in EXP$_{2016-2019}$ during the period of the year of the lockdown (14/03-30/04) is a bit lower but remains reasonable, with a nMB of +4%, a nRMSE of 37% and a PCC of 0.80. Although moderate, such a deterioration of the performance after mid-March might reflect some seasonality in the ML model errors and/or could be related to the presence of trends in the NO2 concentrations. Concerning this last point, as previously discussed in Sect. 2.3.2, including the date index feature in the ML model aims at limiting this potential issue but likely cannot completely solve it. Generally, only minor differences of performance are found between urban background and traffic stations.*

*Results of EXP$_{2020}$ per province (Fig. 2) highlight some inter-regional variability of the performance, with poorer statistics in some provinces, at least for one type of station. At most stations, the bias remains below ±20% while nRMSE ranges between 15 and 45% (highest nRMSE around 50% in Teruel, Tenerife and Fuerteventura). Most provinces show PCC around 0.6-0.9, with only a few exceptions below 0.6 (urban background sites in Bizkaia, Fuerteventura, Huesca and traffic sites in Granada and Gran Canaria)."* Note that we also added a Table A2 in the Appendix with detailed statistics on urban background and traffic stations.

- L159 Figure 1 shows that a number of stations have mean concentrations ~5ppvb. Thus these intervals are very huge, making the result not reliable. I suggest to present these intervals in a different way and not averaging all that data. In this study, the uncertainties affecting our ML predictions are estimated using the most conservative approach, precisely in order to ensure the reliability of the NO2 reductions highlighted. These uncertainty intervals provided are indeed large but correspond to the uncertainties of the ML predictions at the daily scale (between January and April). Therefore, they cannot be compared to the (multi-) annual NO$_2$ averages shown for instance in Figure 1. As already explained in the manuscript, and as expected due to error compensations, the longer the time scale, the shorter these uncertainties. Therefore, the reviewer is here misleading his interpretation of the numbers provided in the text. We modified the sentence to avoid confusion : *"Averaged over all Spanish provinces, the uncertainty interval of ML predictions at the daily*

*scale is [-5.1, +5.3] ppbv at urban background stations, and [-6.6, +6.7] ppbv at traffic stations.*" (Note that the uncertainty intervals are here slightly modified compared to the initial manuscript as they correspond to the results obtained with the extended set of features).

- L167-168 This argument is not clear. Please explain in detail. Here we simply mean that errors at the daily scale can at least partly compensate each other, which implies that averaging the ML-based predictions of daily NO2 mixing ratios to longer time scales (a week for instance) is expected to reduce the uncertainty. This is quite common, also for traditional chemistry transport models (reproducing the daily mean NO2 concentrations always goes with stronger uncertainties than the weekly, monthly or annual mean NO2 concentrations). We modified the sentence: "*These uncertainties are suited for our ML-based daily NO2 predictions. Because these daily uncertainties are likely at least partly uncorrelated, NO2 daily predictions averaged over periods longer than one day are expected to have smaller uncertainties due to error compensations.*"

- Table1 The test cases N seems very low, are these implying number of stations or total number of test days for all stations? Table 1 in the initial version of the manuscript gives the "*the statistics averaged over all Spanish provinces*", so the test cases N corresponds to neither the number of stations, nor the total number of test days, but the number of test days per station (on average over all stations). For each station in each Spanish province, training is performed over 2017-2019 (maximum N for training is therefore 3x365 = 1,095 points per station) and testing over 2020 before lockdown (maximum N for testing is therefore 31+28+14 = 73 points per station). In this Table, statistics were first computed for each station individually, and then averaged together to give the numbers provided in Table 1. Results at individual stations are still visible in Fig. 2.
  In the updated version of the manuscript, we greatly reshaped all this discussion, following a previous comment of the reviewer. Table 1 now gives the overall statistical results, computed over the entire data (i.e. combining all provinces together), which gives a broader view of the performance obtained by the ML-based predictive models.

- L255 In some cities, such zones, resulted in much higher traffic in peripheral road networks. Thus the stations at 3 and 9 km, might experiencing heavier traffic due to LEZ in the center. This should be answered locally by explaining the main routes and the traffic of each city. Investigating in more detail the traffic pattern of Madrid is far beyond the scope of this paper. Although the reviewer is right in principle, to our opinion, the three reasons already mentioned here in the text – namely the very limited area of this LEZ zone (5 km$^2$), the rather large distance to the stations selected and last but not least, the expected progressive transition to a new traffic pattern, given the absence of fines before April 1$^{st}$ (and postponed to September 15$^{th}$ 2020 due to the COVID-19 situation (we added this new element of information in the revised manuscript : "*In our case, we expect a limited impact because the LEZ was still in its transition phase (strict enforcement through fines to offending motorists was not expected until April 1$^{st}$ and was finally postponed to September 15$^{th}$ 2020 due to the COVID-19 situation) and the two stations selected in*

*Madrid province are located outside the LEZ (at 9 and 3 km from the city center).*") – combined together, reasonably justify our assumption that only a "*limited impact is expected*" in Madrid.

- L263 "Statistically significant" should not be used without proper definition and explanation. Explain which significance tests you used, what was the outcome and then provide such conclusions. Here we did not use any statistical test. Uncertainties of daily (weekly) $NO_2$ mixing ratios were computing empirically as the 5th and 95th percentiles of the daily (weekly) residuals obtained over past experiments. They are thus expected (by construction) to represent the 90% confidence interval. We modified the sentence : "*The uncertainty at weekly scale is here used as an estimate of the uncertainty at 90% confidence level (by construction, given that they are computed as the 5th and 95th percentiles of the weekly residuals, see Sect. 2.3.3) affecting the mean NO2 change.*"

- 3.3 I think it is important to present some representative cases of other stations' time series in figures similar to 3 and 4. These provide a very clear picture of the conditions during the lockdown phases. Are there any periods of higher than business as usual concentration, probably in the stations with low mean values (Granada and Murcia probably)? Besides the time series for Madrid and Barcelona (Figs. 3 and 4), we are now providing the Supplement the time series obtained in all other Spanish provinces (Figs. S1-48), in order to allow the reader to check the results obtained in specific locations. Results obtained in the other provinces are generally consistent with those already discussed in Madrid and Barcelona. Thus, we do not think that it is particularly useful to present and discuss other cases in the manuscript.

To answer the specific question of the reviewer, it is indeed possible to encounter observed NO2 concentrations higher to the ML-based business-as-usual concentrations on specific days, although it rarely happens. With the updated results obtained with the extended set of features, over all daily data available during the lockdown, only 4% (110 points over 2844) of the daily NO2 exceed the predicted business-as-usual NO2 estimates. Over these points, the observed NO2 mixing ratios are on average 1.3 ppbv higher than the business-as-usual (20% in relative). For information purpose, we included in the text: "*Results highlight that the reduction previously described in Madrid and Barcelona extends to most Spanish provinces, although with some inter-regional variability in the extent of the change and the degree of statistical significance. During the lockdown period, 96% (2734 points over 2844) of the observed daily NO2 mixing ratios are lower than the ML-based business-as-usual NO2 estimates.*". Note that the corresponding observed NO2 mixing ratios are not particularly low since their average reaches 7.8 ppbv (compared to 5.4 ppbv for the entire NO2 observational dataset). Note also that additional information can already be found in Table 2 where we provided the maximum NO2 changes (among all provinces) during the three different phases and the entire lockdown period : in the revised version of the manuscript, you can see that the maximum NO2 changes (i.e. in our case, the changes closest to zero since values are

negative) are all negative or close to zero (-14% during phases I+II+III for both urban background and traffic stations, -14 and -1% during phase I for urban background and traffic stations, respectively, etc.). This means that although observed NO2 can be higher than the business-as-usual NO2 on specific days, this is never the case along an entire phase (otherwise results would show some increases of NO2 during specific phases).

It is worth noting here that as explained in the manuscript, when selecting the stations, we required at least 10% of daily data during the entire lockdown period (41 days), which represents 4 days. However, we did not apply a similar criteria at the smaller scale of the individual lockdown phases. Although the data coverage in Madrid and Barcelona is very good, in some other provinces, the average NO2 reductions computed during specific lockdown phases can be based on very few data. This can now be seen in the Supplement. If we consider for instance the urban background station in Murcia, data are available during 7, 5 and 5 days in phases I, II and III, respectively (therefore quite well balanced). However, at the urban background station in Granada, data are available during 1, 1 and 9 days in phase I, II and III, respectively. More importantly, the only daily data available in phase I is on the first day of the phase (March 15$^{th}$), i.e. at the very beginning of the lockdown, which likely explains the low increase of NO2 highlighted during phase I (see Fig. A1 in Appendix). The data coverage in these two provinces is almost complete for the traffic station. Over all Spanish provinces, largest data gaps during the lockdown period are found at background stations in Fuerteventura, Granada, Albacete, Alicante, Cuidad Real, Cádiz, Mallorca, Menorca, Murcia and Salamanca, and at traffic stations in Cádiz and Huelva.

We realize now that this can bring some confusion regarding the representativeness of the NO2 reductions highlighted in the paper. Therefore, in the revised version of the manuscript, we now require at least 3 days of available data during each lockdown phase. For computing the NO2 change during phases I+II+III, we required data available during at least 2 over 3 phases, to avoid cases where data is actually available only during one specific phase. As a consequence, some provinces during specific lockdown phases have been removed in Figs. 5 and A1-A4. The overall discussion remains unchanged.

- 3.5 A figure showing all three time series (climatological, business as usual and measured) would be very useful, at least for some representative stations. Following the suggestion of the reviewer, we added the monthly climatological mean NO2 in the time series plots (Figs. 3, 4 and Figs. S1-48 in the Supplement), as well as the NO2 changes obtained with the climatological average approach in Figs. 5 and Figs. A1-A4 in the Appendix.
- L.384-387 This is a very important finding at should be highlighted more and included in the conclusions, because it is general for future application of climatological values. We added the following sentence in the conclusion : "*We also demonstrated the benefits of our meteorology-*

*normalization approach compared to a simple climatological-based approach, especially at smaller temporal and spatial scales.*"

- L445 It is not clear if all the flagged data were removed for the process or if different flags were treated differently. All the flagged data were indeed removed. We added a sentence at the end of this paragraph: "*All the corresponding measurements were removed from the dataset.*"

**Other modifications**
Given the recent publication of a few new relevant studies on the topic (focusing on Spain), we updated some sentences in the manuscript :
- "*While such an extraordinary situation has obviously impacted the levels of air pollution in the country, as seen in both surface and satellite observations (Tobías et al., 2020; Bauwens et al., 2020), the extent of such reductions remains uncertain.*"
- "*Actually, the lockdown offers unique opportunities for so-called dynamical CTM evaluations (Rao et al., 2011), i.e., testing the ability of CTMs to reproduce the observed changes of concentrations under unusually different emissions (Guevara et al., 2020a; Menut et al., 2020).*"
- "*A more detailed analysis of the activity data in these different emission sectors is required to better quantify how the emission forcing has been modified by the lockdown (Guevara et al., 2020a) and to understand the reductions of NO2 obtained in this study.*"
- "*In a separate study, our meteorology-normalized estimates are used to quantify the circumstantial reduction in the mortality attributable to the short-term effects of NO2 during the lockdown (Achebak et al., 2020).*"

With the corresponding references :
- Achebak, H., Petetin, H., Quijal-Zamorano, M., Bowdalo, D., García-Pando, C. P., and Ballester, J.: Reduction in air pollution and attributable mortality due to COVID-19 lockdown, The Lancet Planetary Health, 4, e268, https://doi.org/10.1016/S2542-5196(20)30148-0, https://linkinghub.elsevier.com/retrieve/pii/S2542519620301480, 2020.
- Bauwens, M., Compernolle, S., Stavrakou, T., Müller, J., Gent, J., Eskes, H., Levelt, P. F., van der A, R., Veefkind, J. P., Vlietinck, J., Yu, H., and Zehner, C.: Impact of Coronavirus Outbreak on NO 2 Pollution Assessed Using TROPOMI and OMI Observations, Geophysical Research Letters, 47, https://doi.org/10.1029/2020GL087978, https://onlinelibrary.wiley.com/doi/abs/10.1029/2020GL087978, 2020.
- Guevara, M., Jorba, O., Soret, A., Petetin, H., Bowdalo, D., Serradell, K., Tena, C., Denier van der Gon, H., Kuenen, J., Peuch, V.-H., and Pérez García-Pando, C.: Time-resolved emission reductions for atmospheric chemistry modelling in Europe during the COVID-19 lockdowns (in review), Atmospheric Chemistry and Physics Discussions, https://doi.org/10.5194/acp-2020-686, 2020a
- Menut, L., Bessagnet, B., Siour, G., Mailler, S., Pennel, R., and Cholakian, A.: Impact of lockdown measures to combat Covid-19 on air quality over

western Europe, Science of The Total Environment, 741, 140 426, https://doi.org/10.1016/j.scitotenv.2020.140426, https://linkinghub.elsevier.com/retrieve/pii/S0048969720339486, 2020.

Complete list of changes :
- Title : "*Meteorology-normalized impact of the COVID-19 lockdown upon NO$_2$ pollution in Spain*"
- Affiliations : "*ICREA, Catalan Institution for Research and Advanced Studies, Barcelona, Spain*"
- L1 : "*The spread of the new coronavirus SARS-COV-2 causing COVID-19 forced the Spanish Government [...]*"
- L10 : "*The ML predictive models were found to perform remarkably well in most locations, with overall bias, root-mean-squared error and correlation of +4%, 29% and 0.86, respectively.*"
- L24 : "*The rapid spread of the new coronavirus SARS-COV-2 that causes COVID-19 [...]*"
- L39 : "*While such an extraordinary situation has obviously impacted the levels of air pollution in the country, as seen in both surface and satellite observations (Tobias et al., 2020; Bauwens et al., 2020), the extent of such reductions remains uncertain.*"
- L45 : "*[...] testing the ability of CTMs to reproduce the observed changes of concentrations under unusually different emissions (Guevara et al., 2020b; Menut et al., 2020).*"
- L75 : "*The fraction of E1a data is 0% in 2020, 99% in 2019 and 100% in 2013-2018.*"
- L76 : "*All NO$_2$ measurements taken into account here are operated using chemiluminescence with an internal Molybdenum converter. Although predominantly used over Europe for measuring NO$_2$, this measurement technique is well known to be have strong positive artifacts due to interferences of NO$_z$ compounds (e.g. nitric acid, peroxyacetyl nitrates, organic nitrates), especially during daytime when these species are photo-chemically formed, up to a factor of 2-4 as observed during summertime in urban atmospheres (e.g. Dunlea et al., 2007; Villena et al., 2012). In our case, the positive artifacts at urban background stations are probably lower since the period of study (late winter and early spring) is less photo-chemically active than summertime. Even lower interferences are expected at traffic stations where the NO$_z$/NO$_x$ ratio is typically lower due to the proximity to fresh NOx emissions. In any case, the present study focuses on the relative changes of NO$_2$ due to the lockdown, so biases in the NO$_2$ measurements are of lower importance.*"
- L100 : " *[...] total cloud cover, surface net solar radiation, surface solar radiation downwards, downward UV radiation at the surface and boundary layer height.*"
- L114 : "*Choice of features and modeling strategy*"
- L118 : "*[...] total cloud cover, surface net solar radiation, surface solar radiation downwards, downward UV radiation at the surface, boundary layer height [...]*"

- L124 : "*Including such a feature with unique values (going from 0 for 2013/01/01 to 2669 for 2020/04/23)*"
- L136 : "*This ML experiment is hereafter referred to as EXP$_{2020}$.*"
- L155 : "*These ML experiments are hereafter referred to as EXP$_{2016}$, EXP$_{2017}$, EXP$_{2018}$ and EXP$_{2019}$, respectively.*"
- L159 : "*Averaged over all Spanish provinces, the uncertainty interval is [-5.1, +5.3] ppbv at urban background stations, and [-6.6, +6.7] ppbv at traffic stations.*"
- L167 : "*Because these daily uncertainties are likely at least partly uncorrelated, NO2 daily predictions averaged over time periods longer than one day are expected to have smaller uncertainties due to error compensations.*"
- L172 : "*On average over all provinces, the weekly uncertainty interval obtained are [-3.8, +3.6] ppbv at urban background stations, and [-4.9, +4.7] ppbv at traffic stations, which represents a reduction of 28% for both types of stations, with respect to the daily uncertainties.*"
- L179 : "*Note that these ancillary ML experiments used here for quantifying the uncertainties also allow to evaluate the performance of our modeling strategy during the period of the year of the lockdown (as explained later in Sect. 3.1).*"
- L181 : "*Time series in the other 48 Spanish provinces can be found in the Supplement.*"
- L186 : "*The performance of the ML predictions in each Spanish province and station type is shown in Fig. 2, and the statistics over all Spanish provinces reported in Table 1. Statistical results in Table 1 are given for both the reference ML experiment (EXP2020) and the other experiments combined together (EXP2016, EXP2017, EXP2018 and EXP2019, hereafter referred to as EXP2016–2019). Besides providing a broader view of the performance of our modeling strategy, considering these past experiments also allows assessing the performance of the ML predictions during the period of the year of the lockdown (14/03-30/04, for years 2016 to 2019), which may be important given the potential seasonality of prediction errors. Statistics obtained at urban background and traffic stations are given in Table A2 in Appendix.*"
- L190 : "*For information purposes, we included the statistical results obtained over the training dataset (2017/01/01-2019/12/31 in EXP$_{2020}$). Checking results over the training data may be useful for highlighting obvious situations of overfitting, when the performance is almost perfect. At both urban background and traffic stations, results show no bias, low nRMSE (always below 35%, 19% when considering all provinces), and a high PCC of 0.96. Similar results are obtained when considering the ensemble of all past experiments (EXP$_{2016–2019}$).*"
- L195 : "*On the test dataset of the EXP$_{2020}$ reference experiment (2020/01/01-2020/03/13, before the lockdown), the performance remains reasonably good in most provinces. Over all Spanish provinces, the nMB increases to +4%, the nRMSE to 29% and the PCC is reduced to 0.86, in very close agreement with the performance obtained with EXP$_{2016–2020}$ (nMB of +1%, nRMSE of 28% and PCC of 0.86). In comparison, the performance obtained in EXP$_{2016-2019}$ during the period of the year of the lockdown*"

*(14/03-30/04) is a bit lower but remains reasonable, with a nMB of +4%, a nRMSE of 37% and a PCC of 0.80. Although moderate, such a deterioration of the performance after mid-March might reflect some seasonality in the ML model errors and/or could be related to the presence of trends in the NO2 concentrations. Concerning this last point, as previously discussed in Sect. 2.3.2, including the date index feature in the ML model aims at limiting this potential issue but likely cannot completely solve it. Generally, only minor differences of performance are found between urban background and traffic stations. Results of EXP$_{2020}$ per province (Fig. 2) highlight some inter-regional variability of the performance, with poorer statistics in some provinces, at least for one type of station. At most stations, the bias remains below ±20% while nRMSE ranges between 15 and 45% (highest nRMSE around 50% in Teruel, Tenerife and Fuerteventura). Most provinces show PCC around 0.6-0.9, with only a few exceptions below 0.6 (urban background sites in Bizkaia, Fuerteventura, Huesca and traffic sites in Granada and Gran Canaria)."*

- L225 : *"like in the Canary Islands (e.g. Tenerife and Fuerteventura)."*
- L233 : *"89% (4240 points over 4788)"*
- L246 : *"(nMB of -3 and +6%, nRMSE of 19 and 22%, PCC of 0.87 and 0.85, respectively)."*
- L254 : *"(strict enforcement through fines to offending motorists was not expected until April 1st and was finally postponed to September 15th 2020 due to the COVID-19 situation)"*
- L265 : *"The uncertainty at weekly scale is here used as an estimate of the uncertainty at 90% confidence level (by construction, given that they are computed as the 5th and 95th percentiles of the weekly residuals, see Sect. 2.3.3) affecting the mean NO$_2$ change."*
- L267 : *"-7[-13,-1] ppbv"*
- L268 : *"-39[-74,-4]%"*
- L269 : *"-10[-15,-5] ppbv, or -59[-87,-30]%"*
- L276 : *"(nRMSE of 25%) and correlations (PCC of 0.72)"*
- L276 : *"The positive bias in the traffic station started in early February and persisted during the following weeks"*
- L277 : *"(+0%), and reaches +8%"*
- L284 : *"start before April 1$_{st}$ (postponed to September 15$_{th}$ 2020 due to the COVID-19 situation)."*
- L304 : *"decreased by -7[-12,-2] ppbv (-47[-78,-16]%)"*
- L306 : *"-15[-20,-10] ppbv (-61[-80,-38]%)."*
- L317 : *"significance. During the lockdown period, 96% (2734 points over 2844) of the observed daily NO2 mixing ratios are lower than the ML-based business-as-usual NO2 estimates."*
- L318 : *"-4[-8,-0] ppbv (-49[-95,-0]% in relative terms)"*
- L320 : *"and -1 ppbv (-31%)."*
- L321 : *"22 out of 38 provinces,"*
- L322 : *"-7[-11,-2] ppbv (or -50[-91,-8]%), and 26 out of 37 stations"*
- L329 : *"about -42% at both station types, and further increased to about -54% during phases II and III."*

- L332 : *"between -20 and -40% depending on the type of station during phases II and III, compared to only -9 to -19% during phase I."*
- L337 : *"Barcelona Supercomputing Center (Guevara et al., 2020b)."*
- L353 : *"lockdown (Guevara et al., 2020a)"*
- L364 : *"-44 and -53% at the urban background and traffic stations, respectively"*
- L367 : *"-50 and -63% at urban background and traffic stations"*
- L368 : *"NO2 reductions of -43 and -60%"*
- L382 : *"The NO2 changes obtained with the climatological average approach are reported on Fig. 5 (and for the different phases in Figs. A1, A2, A3, A4 in Appendix)."*
- L391 : *"biased by +27%."*
- L395 : *"+12, +2.3 and +1.8%"*
- L396 : *"-21/+52, -34/+44 and -41/+36% during phases I, II and III, respectively. For the case of Barcelona province, these relative biases are +35, +19 and 22%."*
- L412 : *"fed by meteorological data and time variables (Julian date, day of week and date index)"*
- L417 : *"We also demonstrated the benefits of our meteorology-normalization approach compared to a simple climatological-based approach, especially at smaller temporal and spatial scales."*
- L440 : *"The results of the present study provide a valuable reference for validating similar assessments of the impact of the COVID-19 lockdown on air quality based on chemistry transport models and emission scenarios derived from activity data during the lockdown (e.g. Guevara et al., 2020a; Menut et al., 2020)."*
- L441 : *"during the lockdown (Achebak et al., 2020)."*
- L442 : *"EEA AQ e-Reporting,"*
- L458 : *"All the corresponding measurements were removed from the dataset."*

Figures and tables :
- We modified the color bar of Figs 1 and 6
- We reshaped Table 1 and its legend
- We added monthly climatological NO2 mixing ratios on Figs. 3 and 4, and modified the legend : *"The climatological monthly averages computed over the period 2017-2019 are also shown (in black). The vertical black line identifies the beginning of the lockdown, the next dotted lines separate the different lockdown phases (phase I : 2020/03/14-2020/03/29, phase II : 2020/03/30-2020/04/09, phase III : 2020/04/10-2020/04/23)."*
- NO2 changes in Table 2 have been slightly modified, according to the new results obtained with the extended set of features.
- We added the NO2 changes obtained with the climatological average approach in Fig. 5 and modified the legend : *"For comparison, the mean NO2 changes obtained using the climatological average (over 2017-2019) rather than ML-based business-as-usual NO2 concentration are also shown (stars), as well as the relative difference between both approaches (circles)."*

Appendix :
- Figs A1-A4 have been modified (we added information regarding NO2 changes obtained with the climatological average approach)
- Table A2 added (with detailed information about the statistical results obtained at urban background and traffic stations)

Supplement : We included the time series (similar to Figs. 3 and 4) for 48 Spanish provinces.

---

## Author Comment (AC2) · 31 Aug 2020

**Reviewer #1**

Comments from the reviewer are in blue, and answers in black (text citations and modifications are highlighted in italics). Note that following the recommendation of the other reviewer, we added three new meteorological features (surface net solar radiation, surface solar radiation downwards, downward UV radiation at the surface) and updated all the figures, tables and corresponding text. The impact on the results is relatively small so the discussion remains essentially the same.

The article under review here aims to quantify the impact of the Covid-19 lockdown measures in Spain on air quality. The topic is interesting from the point of view of air quality practitioners and the general public, but it also raises substantial scientific challenges. Even if economic activities were substantially reduced during the lock down period, the impact of meteorological factors on air quality precludes a simple comparison with previous years. Instead, the authors mobilize innovative machine learning approaches to tackle the issue. The quality of the presentation, scientific quality, and societal relevance are excellent, and publication in ACP is therefore recommended. I am nevertheless proposing the following minor suggestions that could help further strengthen the paper.

We are thankful to the reviewer for his/her positive feedbacks and comments.

General comment:
The authors should be encouraged to extent the coverage of their study. Applying the method over the whole of Europe is certainly the scope for another paper. But an extension of the temporal coverage up to the end of the lockdown in Spain would be interesting.

We agree that an extension over Europe is interesting, and we are currently collaborating on another study addressing the question at this larger scale (focusing on the largest European cities). Concerning the extension of the temporal coverage of the present study, we took into account the time period with data available at the time of preparation/submission of this study. Although it would have been nice to cover the entire period of the lockdown, we are here considering a period already quite extended (41 days), comprising the most stringent phase of the lockdown. To our opinion, although interesting, extending the study would require to substantially reshape the first draft, without bringing much more scientific knowledge. In addition, even at the time of this revision (August 25th), the situation cannot be considered as normal since many people across Spain are still working from home in Spain (and some parts of the country have been recently confined again).

Specific comments:
- L24, L403: the coronavirus is SARS-COV-2 not COVID-19. Indeed, the reviewer is right, according to the World Health Organization, COVID-19 designates the coronavirus disease, while SARS-COV-2 refers to the virus itself. To be consistent with this terminology, we added the term "*disease*" in the text.
- L36: without supporting reference, it is wiser to state that "the impact on industry is *presumably* more contrasted". Corrected.

- L50: in the motivation of the work, the authors could add that this type of analysis will serve to validate the model-based assessment using emission scenarios derived from activity data during the lockdown. We added in the conclusion : "*The results of the present study provide a valuable reference for validating similar assessments of the impact of the COVID-19 lockdown on air quality based on chemistry transport models and emission scenarios derived from activity data during the lockdown (e.g. Guevara et al., 2020a; Menut et al., 2020).*" with the corresponding references :
    - Menut, L., Bessagnet, B., Siour, G., Mailler, S., Pennel, R., and Cholakian, A.: Impact of lockdown measures to combat Covid-19 on air quality over western Europe, Science of The Total Environment, 741, 140 426, https://doi.org/10.1016/j.scitotenv.2020.140426, https://linkinghub.elsevier.com/retrieve/pii/S0048969720339486, 2020.
    - Guevara, M., Jorba, O., Soret, A., Petetin, H., Bowdalo, D., Serradell, K., Tena, C., Denier van der Gon, H., Kuenen, J., Peuch, V.-H., and Pérez García-Pando, C.: Time-resolved emission reductions for atmospheric chemistry modelling in Europe during the COVID-19 lockdowns (in review), Atmospheric Chemistry and Physics Discussions, https://doi.org/10.5194/acp-2020-686, 2020a.
- L69: where is the GHOST data available ? If GHOST database is not publicly open, the reference of the availability of the data should remain EEA's AQ e-reporting database. GHOST is a BSC internal on-going project currently not publicly available and a publication describing the dataset is in preparation. As explained in the text, GHOST is not another database, it ingests different air quality publicly available databases (including the EEA AQ eReporting database used in this study) and provides consistent and extended metadata to ensure the quality of the observational data. Although neglected by many studies, we consider that this quality assurance screening is an essential part of the data preprocessing. This is why we consider that it is worth mentioning and explaining it in detail in the manuscript, while to our opinion, the reference to the use of the EEA AQ eReporting database is already clear enough in the text.
- L75: the formal deadline for 2019 AQ e-reporting data to be delivered as E1a is September 2020, what is the fraction of 2019 data already E1a at the date of submission? Regarding the September deadline, it seems that many countries are actually delivering E1a data earlier (sometimes bit by bit through the year). We added the following text : "*The fraction of E1a data is 0% in 2020, 99% in 2019 and 100% in 2013-2018.*"
- L125: please clarify what you mean by "unique values", is the date index the Julian day, and if so why would it be unique? There is here a misunderstanding. As explained in L119, the date index is the number of days since 2013/01/01 (i.e. unique values going from 0 for 2013/01/01 to 2677 for 2020/04/30), while the Julian date (going from 1 to 365) is another feature. We added this to the sentence : "*Including such a feature with unique values (going from 0 for 2013/01/01 to 2677 for 2020/04/30) is not expected* [...]"

- L145: hyperparameters should be defined and discussed either in the main text or in the annex. Further details would be appreciated in the annex on how the choice of those hyperparameters are related with the problem at hand (density and spread of observations, number and diversity of predictors etc.). The tuning strategy is explained in detail in Appendix C. The hyperparameters selected here are very common to any ML exercise with the gradient boosting machine and are not tailored to our specific problem. For each of these hyperparameters, we defined a reasonably large range of possible values to be tested through a randomized search, following again the idea we have about the common practices in the field (and the computational resources available for these calculations). We are not arguing here that this tuning strategy optimizes the best the performance but the performance obtained was found to be acceptable for the present study.

- L245: include the value of the uncertainty interval, it is difficult to compare percentages in 3.2 and ppbv intervals in 2.3.3. Actually, both should not be compared because they are not directly comparable. There is here a misunderstanding since the uncertainty intervals of Sect. 2.3.3 correspond to the uncertainties of the ML predictions at the daily and weekly scales (i.e. the uncertainties of the daily or weekly average $NO_2$ concentrations).

- L255: the impact of the LEZ could actually be an increase of $NO_2$ at stations in the outskirts of that zone. As also explained in our answer to the first reviewer, although the reviewer is right in principle, to our opinion, the 3 reasons already mentioned here in the text (namely the very limited area of this LEZ zone (5 km$^2$), the rather large distance to the stations selected and last but not least, the expected progressive transition to a new traffic pattern, given the absence of fines before April 1$^{st}$, now postponed to September 15$^{th}$ 2020), combined together, reasonably justify our assumption that only a "*limited impact is expected*" in Madrid.

- Figure 2: N seems to be missing from the plot. Thanks, we corrected it (this was an old version of the legend).

- L266: clarify if the confidence interval is taken from the distribution of daily differences. We are not sure to properly understand what should be clarified here. The uncertainties used here correspond to the uncertainties at weekly scale (computed based on the differences between $NO_2$ observations and predictions weekly averaged, as explained in Sect. 2.3.3). If the reviewer is talking about the uncertainties at daily scale, they are indeed obtained from the distribution of the daily differences.

- L325 and L344: could there be a role of background ozone in the relation between NOx emission changes and $NO_2$ concentrations that would appear through this latitudinal gradient? The $NO_2$ reductions obtained tend to be stronger in the southern half of Spain, but there is not a very clear latitudinal gradient that apply to all provinces. For instance, relatively lower $NO_2$ reductions are found along the southern coast of Spain. Ozone and other chemical compounds may in principle impact the

NO2 concentrations (directly or indirectly) but we do not have any clear evidence for this at this stage.

- L365: clarify which reduction is for urban and traffic stations. We modified the text as follows : "*On average over this set of provinces, the NO2 reduction is -44 and -53% at the urban background and traffic stations, respectively* [...]"
- L412: also mention day of the week in the predictors, which is presumably very important for NO2. We modified the sentence as follows : "*To tackle this issue, we used ML models fed by meteorological data and time variables (Julian date, day of week and date index) to estimate* [...]"

**Other modifications**

Given the recent publication of a few new relevant studies on the topic (focusing on Spain), we updated some sentences in the manuscript :

- "*While such an extraordinary situation has obviously impacted the levels of air pollution in the country, as seen in both surface and satellite observations (Tobías et al., 2020; Bauwens et al., 2020), the extent of such reductions remains uncertain.*"
- "*Actually, the lockdown offers unique opportunities for so-called dynamical CTM evaluations (Rao et al., 2011), i.e., testing the ability of CTMs to reproduce the observed changes of concentrations under unusually different emissions (Guevara et al., 2020a; Menut et al., 2020).*"
- "*A more detailed analysis of the activity data in these different emission sectors is required to better quantify how the emission forcing has been modified by the lockdown (Guevara et al., 2020a) and to understand the reductions of NO2 obtained in this study.*"
- "*In a separate study, our meteorology-normalized estimates are used to quantify the circumstantial reduction in the mortality attributable to the short-term effects of NO2 during the lockdown (Achebak et al., 2020).*"

With the corresponding references :

- Achebak, H., Petetin, H., Quijal-Zamorano, M., Bowdalo, D., García-Pando, C. P., and Ballester, J.: Reduction in air pollution and attributable mortality due to COVID-19 lockdown, The Lancet Planetary Health, 4, e268, https://doi.org/10.1016/S2542-5196(20)30148-0, https://linkinghub.elsevier.com/retrieve/pii/S2542519620301480, 2020.
- Bauwens, M., Compernolle, S., Stavrakou, T., Müller, J., Gent, J., Eskes, H., Levelt, P. F., van der A, R., Veefkind, J. P., Vlietinck, J., Yu, H., and Zehner, C.: Impact of Coronavirus Outbreak on NO 2 Pollution Assessed Using TROPOMI and OMI Observations, Geophysical Research Letters, 47, https://doi.org/10.1029/2020GL087978, https://onlinelibrary.wiley.com/doi/abs/10.1029/2020GL087978, 2020.
- Guevara, M., Jorba, O., Soret, A., Petetin, H., Bowdalo, D., Serradell, K., Tena, C., Denier van der Gon, H., Kuenen, J., Peuch, V.-H., and Pérez García-Pando, C.: Time-resolved emission reductions for atmospheric chemistry modelling in Europe during the COVID-19 lockdowns (in review),

Atmospheric Chemistry and Physics Discussions,
https://doi.org/10.5194/acp-2020-686, 2020a

- Menut, L., Bessagnet, B., Siour, G., Mailler, S., Pennel, R., and Cholakian, A.: Impact of lockdown measures to combat Covid-19 on air quality over western Europe, Science of The Total Environment, 741, 140 426, https://doi.org/10.1016/j.scitotenv.2020.140426, https://linkinghub.elsevier.com/retrieve/pii/S0048969720339486, 2020.

Complete list of changes :

- Title : "*Meteorology-normalized impact of the COVID-19 lockdown upon NO$_2$ pollution in Spain*"
- Affiliations : "*ICREA, Catalan Institution for Research and Advanced Studies, Barcelona, Spain*"
- L1 : "*The spread of the new coronavirus SARS-COV-2 causing COVID-19 forced the Spanish Government [...]*"
- L10 : "*The ML predictive models were found to perform remarkably well in most locations, with overall bias, root-mean-squared error and correlation of +4%, 29% and 0.86, respectively.*"
- L24 : "*The rapid spread of the new coronavirus SARS-COV-2 that causes COVID-19 [...]*"
- L39 : "*While such an extraordinary situation has obviously impacted the levels of air pollution in the country, as seen in both surface and satellite observations (Tobias et al., 2020; Bauwens et al., 2020), the extent of such reductions remains uncertain.*"
- L45 : "*[...] testing the ability of CTMs to reproduce the observed changes of concentrations under unusually different emissions (Guevara et al., 2020b; Menut et al., 2020).*"
- L75 : "*The fraction of E1a data is 0% in 2020, 99% in 2019 and 100% in 2013-2018.*"
- L76 : "*All NO$_2$ measurements taken into account here are operated using chemiluminescence with an internal Molybdenum converter. Although predominantly used over Europe for measuring NO$_2$, this measurement technique is well known to be have strong positive artifacts due to interferences of NO$_z$ compounds (e.g. nitric acid, peroxyacetyl nitrates, organic nitrates), especially during daytime when these species are photo-chemically formed, up to a factor of 2-4 as observed during summertime in urban atmospheres (e.g. Dunlea et al., 2007; Villena et al., 2012). In our case, the positive artifacts at urban background stations are probably lower since the period of study (late winter and early spring) is less photo-chemically active than summertime. Even lower interferences are expected at traffic stations where the NO$_z$/NO$_x$ ratio is typically lower due to the proximity to fresh NOx emissions. In any case, the present study focuses on the relative changes of NO$_2$ due to the lockdown, so biases in the NO$_2$ measurements are of lower importance.*"
- L100 : " *[...] total cloud cover, surface net solar radiation, surface solar radiation downwards, downward UV radiation at the surface and boundary layer height.*"
- L114 : "*Choice of features and modeling strategy*"

- L118 : *"[…] total cloud cover, surface net solar radiation, surface solar radiation downwards, downward UV radiation at the surface, boundary layer height […]"*
- L124 : *"Including such a feature with unique values (going from 0 for 2013/01/01 to 2669 for 2020/04/23)"*
- L136 : *"This ML experiment is hereafter referred to as EXP$_{2020}$."*
- L155 : *"These ML experiments are hereafter referred to as EXP$_{2016}$, EXP$_{2017}$, EXP$_{2018}$ and EXP$_{2019}$, respectively."*
- L159 : *"Averaged over all Spanish provinces, the uncertainty interval is [-5.1, +5.3] ppbv at urban background stations, and [-6.6, +6.7] ppbv at traffic stations."*
- L167 : *"Because these daily uncertainties are likely at least partly uncorrelated, NO2 daily predictions averaged over time periods longer than one day are expected to have smaller uncertainties due to error compensations."*
- L172 : *"On average over all provinces, the weekly uncertainty interval obtained are [-3.8, +3.6] ppbv at urban background stations, and [-4.9, +4.7] ppbv at traffic stations, which represents a reduction of 28% for both types of stations, with respect to the daily uncertainties."*
- L179 : *"Note that these ancillary ML experiments used here for quantifying the uncertainties also allow to evaluate the performance of our modeling strategy during the period of the year of the lockdown (as explained later in Sect. 3.1)."*
- L181 : *"Time series in the other 48 Spanish provinces can be found in the Supplement."*
- L186 : *"The performance of the ML predictions in each Spanish province and station type is shown in Fig. 2, and the statistics over all Spanish provinces reported in Table 1. Statistical results in Table 1 are given for both the reference ML experiment (EXP2020) and the other experiments combined together (EXP2016, EXP2017, EXP2018 and EXP2019, hereafter referred to as EXP2016–2019). Besides providing a broader view of the performance of our modeling strategy, considering these past experiments also allows assessing the performance of the ML predictions during the period of the year of the lockdown (14/03-30/04, for years 2016 to 2019), which may be important given the potential seasonality of prediction errors. Statistics obtained at urban background and traffic stations are given in Table A2 in Appendix."*
- L190 : *"For information purposes, we included the statistical results obtained over the training dataset (2017/01/01-2019/12/31 in EXP$_{2020}$). Checking results over the training data may be useful for highlighting obvious situations of overfitting, when the performance is almost perfect. At both urban background and traffic stations, results show no bias, low nRMSE (always below 35%, 19% when considering all provinces), and a high PCC of 0.96. Similar results are obtained when considering the ensemble of all past experiments (EXP$_{2016-2019}$)."*
- L195 : *"On the test dataset of the EXP$_{2020}$ reference experiment (2020/01/01-2020/03/13, before the lockdown), the performance remains reasonably good in most provinces. Over all Spanish provinces, the nMB increases to +4%, the nRMSE to 29% and the PCC is reduced to 0.86, in*

*very close agreement with the performance obtained with EXP$_{2016-2020}$ (nMB of +1%, nRMSE of 28% and PCC of 0.86). In comparison, the performance obtained in EXP$_{2016-2019}$ during the period of the year of the lockdown (14/03-30/04) is a bit lower but remains reasonable, with a nMB of +4%, a nRMSE of 37% and a PCC of 0.80. Although moderate, such a deterioration of the performance after mid-March might reflect some seasonality in the ML model errors and/or could be related to the presence of trends in the NO2 concentrations. Concerning this last point, as previously discussed in Sect. 2.3.2, including the date index feature in the ML model aims at limiting this potential issue but likely cannot completely solve it. Generally, only minor differences of performance are found between urban background and traffic stations. Results of EXP$_{2020}$ per province (Fig. 2) highlight some inter-regional variability of the performance, with poorer statistics in some provinces, at least for one type of station. At most stations, the bias remains below ±20% while nRMSE ranges between 15 and 45% (highest nRMSE around 50% in Teruel, Tenerife and Fuerteventura). Most provinces show PCC around 0.6-0.9, with only a few exceptions below 0.6 (urban background sites in Bizkaia, Fuerteventura, Huesca and traffic sites in Granada and Gran Canaria)."*

- L225 : *"like in the Canary Islands (e.g. Tenerife and Fuerteventura)."*
- L233 : *"89% (4240 points over 4788)"*
- L246 : *"(nMB of -3 and +6%, nRMSE of 19 and 22%, PCC of 0.87 and 0.85, respectively)."*
- L254 : *"(strict enforcement through fines to offending motorists was not expected until April 1st and was finally postponed to September 15th 2020 due to the COVID-19 situation)"*
- L265 : *"The uncertainty at weekly scale is here used as an estimate of the uncertainty at 90% confidence level (by construction, given that they are computed as the 5th and 95th percentiles of the weekly residuals, see Sect. 2.3.3) affecting the mean NO$_2$ change."*
- L267 : *"-7[-13,-1] ppbv"*
- L268 : *"-39[-74,-4]%"*
- L269 : *"-10[-15,-5] ppbv, or -59[-87,-30]%"*
- L276 : *"(nRMSE of 25%) and correlations (PCC of 0.72)"*
- L276 : *"The positive bias in the traffic station started in early February and persisted during the following weeks"*
- L277 : *"(+0%), and reaches +8%"*
- L284 : *"start before April 1$_{st}$ (postponed to September 15$_{th}$ 2020 due to the COVID-19 situation)."*
- L304 : *"decreased by -7[-12,-2] ppbv (-47[-78,-16]%)"*
- L306 : *"-15[-20,-10] ppbv (-61[-80,-38]%)."*
- L317 : *"significance. During the lockdown period, 96% (2734 points over 2844) of the observed daily NO2 mixing ratios are lower than the ML-based business-as-usual NO2 estimates."*
- L318 : *"-4[-8,-0] ppbv (-49[-95,-0]% in relative terms)"*
- L320 : *"and -1 ppbv (-31%)."*
- L321 : *"22 out of 38 provinces,"*
- L322 : *"-7[-11,-2] ppbv (or -50[-91,-8]%), and 26 out of 37 stations"*

- L329 : *"about -42% at both station types, and further increased to about -54% during phases II and III."*
- L332 : *"between -20 and -40% depending on the type of station during phases II and III, compared to only -9 to -19% during phase I."*
- L337 : *"Barcelona Supercomputing Center (Guevara et al., 2020b)."*
- L353 : *"lockdown (Guevara et al., 2020a)"*
- L364 : *"-44 and -53% at the urban background and traffic stations, respectively"*
- L367 : *"-50 and -63% at urban background and traffic stations"*
- L368 : *"NO2 reductions of -43 and -60%"*
- L382 : *"The NO2 changes obtained with the climatological average approach are reported on Fig. 5 (and for the different phases in Figs. A1, A2, A3, A4 in Appendix)."*
- L391 : *"biased by +27%."*
- L395 : *"+12, +2.3 and +1.8%"*
- L396 : *"-21/+52, -34/+44 and -41/+36% during phases I, II and III, respectively. For the case of Barcelona province, these relative biases are +35, +19 and 22%."*
- L412 : *"fed by meteorological data and time variables (Julian date, day of week and date index)"*
- L417 : *"We also demonstrated the benefits of our meteorology-normalization approach compared to a simple climatological-based approach, especially at smaller temporal and spatial scales."*
- L440 : *"The results of the present study provide a valuable reference for validating similar assessments of the impact of the COVID-19 lockdown on air quality based on chemistry transport models and emission scenarios derived from activity data during the lockdown (e.g. Guevara et al., 2020a; Menut et al., 2020)."*
- L441 : *"during the lockdown (Achebak et al., 2020)."*
- L442 : *"EEA AQ e-Reporting,"*
- L458 : *"All the corresponding measurements were removed from the dataset."*

Figures and tables :
- We modified the color bar of Figs 1 and 6
- We reshaped Table 1 and its legend
- We added monthly climatological NO2 mixing ratios on Figs. 3 and 4, and modified the legend : *"The climatological monthly averages computed over the period 2017-2019 are also shown (in black). The vertical black line identifies the beginning of the lockdown, the next dotted lines separate the different lockdown phases (phase I : 2020/03/14-2020/03/29, phase II : 2020/03/30-2020/04/09, phase III : 2020/04/10-2020/04/23)."*
- NO2 changes in Table 2 have been slightly modified, according to the new results obtained with the extended set of features.
- We added the NO2 changes obtained with the climatological average approach in Fig. 5 and modified the legend : *"For comparison, the mean NO$_2$ changes obtained using the climatological average (over 2017-2019)*

*rather than ML-based business-as-usual NO2 concentration are also shown (stars), as well as the relative difference between both approaches (circles)."*

Appendix :
- Figs A1-A4 have been modified (we added information regarding NO2 changes obtained with the climatological average approach)
- Table A2 added (with detailed information about the statistical results obtained at urban background and traffic stations)

Supplement : We included the time series (similar to Figs. 3 and 4) for 48 Spanish provinces.